# Towards Understanding the Effect of NTP Paradigm in Unstructured Knowledge Editing

## Abstract

Editing Large language models (LLMs) with real-world, unstructured knowledge is critical for correcting and updating their internal knowledge bases. However, current methods often oversimplify this knowledge, leading to information loss and suboptimal performance. While existing editing techniques based on the next-token prediction (NTP) paradigm show promise, our investigation reveal a core limitation: **context reliance**. The edited knowledge heavily rely on the preceding context available during editing, but this context is often absent in practical inference. This gap between editing and inference limits the generalization of acquired knowledge. We validate this issue both theoretically and experimentally, demonstrating that the absence of preceding context prevents model from recalling the edited knowledge, thereby causing a performance drop on editing success rate. To address this, we propose a simple yet effective **CO**ntext-**IN**dependent unstructured knowledge editing framework (COIN), encouraging the model to internalize new knowledge properly, rather than merely memorizing fixed patterns with its preceding context. Comprehensive evaluations show that COIN significantly reduces the performance drop and outperforms strong baselines by 23.6% in editing success rate, highlighting the potential of NTP paradigm for robust unstructured knowledge editing.

## 1 Introduction

Large language models (LLMs) encode extensive world knowledge during their pre-training phase (Zhao et al., 2023). However, this acquired knowledge is static and can contain factual inaccuracies or become outdated over time (De Cao et al., 2021). Knowledge editing (Yao et al., 2023) has emerged as a promising technique to address this limitation, which enables efficient and precise modification to specific knowledge within the model. While most existing research has focused on structured knowledge editing, typically modifying factual triplets in a (*subject*, *relation*, *object*) format (Meng et al., 2022; Zhang et al., 2024; Fang et al., 2025), the vast majority of real-world knowledge is stored as unstructured, free-form text (Bavota, 2016). Such text is far more complex, characterized by richer semantic detail, long-form content, and intricate logical relationships, making it more challenging to edit. The goal of unstructured knowledge editing is to edit the model utilizing a text, enabling it to reason based on the newly incorporated information. Therefore, extending editing techniques to effectively handle the unstructured is crucial for practical application.

Current unstructured knowledge editing methods often oversimplify this complex knowledge, leading to the loss of critical information. As illustrated in Figure 1 (a), these methods primarily employ two strategies. The first approach decomposes unstructured knowledge into a series of triplets, which are then edited utilizing editing techniques for the structured (Liu et al., 2024). However, this process can lead to the loss of semantic connections and logical relationships within the original text. The second approach involves generating a query for the target text and editing the corresponding query-answer (QA) pair (Deng et al., 2024; Jiang et al., 2025). This strategy is also often ineffective, as a single query is typically insufficient to capture the full scope of knowledge embedded in complex text. Their suboptimal performance shown in Table 1 further highlights these limitations.

The standard next-token prediction (NTP) paradigm utilized in pre-training enables LLMs to acquire knowledge naturally from unstructured text (Xiong et al., 2025). While this approach theoretically well-suited for direct editing the full text and preventing information loss, its efficacy for

Figure 1: Comparison of different editing methods. (a) illustrates that current approaches over-simplify unstructured knowledge editing to the structured editing, leading to information loss. (b) illustrates that while the NTP paradigm avoids this loss by editing the full text, we observe a performance decline, and attribute this to context reliance, a phenomenon where the newly acquired knowledge is excessively dependent on its preceding context.

unstructured knowledge editing has not been thoroughly examined. To bridge this gap, we conducted experiments on serval unstructured knowledge editing benchmarks to get deeper insights into this paradigm. Our initial investigation uncovers a critical limitation, where the model's accuracy in answering questions degrades when the relevant knowledge is located towards the end of the input text, as shown in Figure 1 (b). We attribute this performance decline to **context reliance**, where knowledge acquired from edited texts relies heavily on its preceding context during editing. Consequently, when queried in isolation without that context during testing, the model generates inaccurate responses. We validate this issue both theoretically and experimentally, demonstrating that the absence of the original context hinders the model's ability to recall the edited knowledge during inference. For example, after a model learns the fact that "*Messi won the World Cup in 2022.*" from a text that also includes the preceding context "*Messi was born on 24 June 1987 . . . ,*" it may associate this fact with the context. If this context is omitted when asking, "*What did Messi won in 2022?*", the model may fail to recall the fact about the "*World Cup*". These findings highlight that the standard NTP paradigm struggles to perform effective and reliable unstructured knowledge editing in realistic scenarios.

Motivated by this, we propose a simple yet effective **CO**ntext-**IN**dependent unstructured knowledge editing framework (COIN), requiring model to internalize new knowledge rather than simply associating it with preceding context when editing. Specifically, COIN introduces the context alignment loss aligning knowledge distribution conditioned on contexts of varying lengths. This objective forces model to concentrate on the critical information contained in local context, thereby reducing the reliance of the acquired knowledge on its entire context and ensuring the generalization of knowledge in practical downstream scenarios. To prevent model collapse (Yang et al., 2024b), we also propose the knowledge consistency loss, ensuring the model's behavior on unrelated knowledge inputs remains consistent before and after edits. To validate the efficacy of COIN, we conduct comprehensive evaluation using two LLMs: Llama3-8B (Grattafiori et al., 2024) and Qwen2.5-7B (Yang et al., 2024a). The evaluation covers both unstructured and structured knowledge editing tasks. For unstructured knowledge editing, COIN reduces the performance drop by 45.2%, and achieves over a 25.6% improvement in editing success rate on benchmarks including AKEW (Wu et al., 2024) and UnKEBench (Deng et al., 2024). For structured knowledge, it outperforms all baseline models on multi-hop task, demonstrating superior generalization capabilities. These results suggest that the editing methods based on NTP paradigm hold significant promise for real-world applications.

We summarize our contributions as follows:

• We theoretically and empirically identify context reliance as a key obstacle to unstructured knowledge editing based on the NTP paradigm, where knowledge learned from edited texts is overly dependent on its preceding context.

• We propose COIN, a simple yet effective framework for context-independent unstructured knowledge editing. COIN guides the edited model to internalize new knowledge rather than associating it with the preceding context.

• We conduct comprehensive experiments on both unstructured and structured knowledge editing benchmarks. The results demonstrate that COIN significantly alleviates the context reliance is-

sue and outperforms existing SOTA methods in editing accuracy, highlighting its potential for practical, real-world knowledge editing paradigm.

## 2 RELATED WORK

**Structured Knowledge Editing** addresses well-defined, knowledge represented as (*subject*, *relation*, *object*) triplets. These methods can be broadly classified into three categories: external memorization-based (Hartvigsen et al., 2023; Huang et al., 2023), eeta-Learning-based (Mitchell et al., 2022a; Tan et al., 2024), and locate-then-edit methods (Dai et al., 2022; Fang et al., 2025).

**Unstructured Knowledge Editing** methods are designed for more realistic scenarios where knowledge is in complex texts. These methods can be categorized into two groups. **Triplets decomposition-based** methods decompose texts into triplets, then apply structured knowledge editing techniques. Liu et al. (2024) utilize eventual context to decompose text into subquestions and corresponding answers to perform editing. **Query construction-based** methods focus on constructing a query for target text, then edit model based on the QA pair. UnKE (Deng et al., 2024) collects context information in input query across multiple layers and utilizes it to inject knowledge into specific MLP module. AnyEdit (Jiang et al., 2025) breaks down unstructured knowledge into blocks, editing them based on the query iteratively. More detailed related work is provided in Appendix D.

Despite these advancements, current unstructured knowledge editing methods often oversimplify complex text, leading to the loss of critical information. To address this, we investigate unstructured knowledge editing based on the NTP paradigm, which directly utilizes raw knowledge text for editing, theoretically capable of effectively handling unstructured knowledge.

## 3 ANALYSIS ON NEXT-TOKEN PREDICTION

In this section, we first examine the performance of the NTP editing paradigm in handling unstructured knowledge, observing a performance decline as knowledge appears later in the text. We then analyze this issue, attributing it to the knowledge's dependence on preceding context, and provide validation for this observation. Finally, we evaluate the effectiveness of existing techniques in mitigating this new issue.

### 3.1 EXPERIMENTAL SETUP

**Models & Methods.** For our experiments, we select the pretrained and instruction-tuned versions of Llama3-8B and Qwen2.5-7B models. We evaluate three training methods, including Fine-Tuning (FT) (Rawat et al., 2021), LoRA (Hu et al., 2022), and AdaLoRA (Zhang et al., 2023). Specifically, all three methods employ NTP paradigm for training on entire unstructured text. FT optimize particular MLP module, while LoRA and AdaLoRA introduce low-rank adapters to the weight matrices of each MLP layer for training. Further details on these methods are provided in Appendix F.

**Dataset.** To evaluate the performance of the NTP paradigm in editing unstructured knowledge, we utilize the AKEW dataset (Wu et al., 2024). Each instance in AKEW contains an unstructured text and several questions formatted as text completion tasks, where each question targets a specific piece of knowledge found within that text. Crucially, these questions are ordered sequentially based on the location of the targeting knowledge within the original text (i.e., the first question targets knowledge appearing at the beginning, whereas subsequent questions target knowledge appearing later). This ordering enables a granular analysis of how preceding context affects editing performance. We apply the NTP editing method directly to the original unstructured text without any pre-processing. After editing, we measure performance by assessing the model's ability to answer questions targeting specific knowledge, rather than by having it reproduce the entire text. To investigate the impact of data format, we also employed a LLM to convert the original text completion tasks into a standard QA format. Further details about the AKEW dataset are available in Appendix E.1.

**Metrics.** For a comprehensive performance evaluation, following Deng et al. (2024) and Jiang et al. (2025), we adopt BERT and ROUGE Scores as our primary metrics. Specifically, we employ BERT Score to evaluate the semantic similarity between the model-generated output and the ground

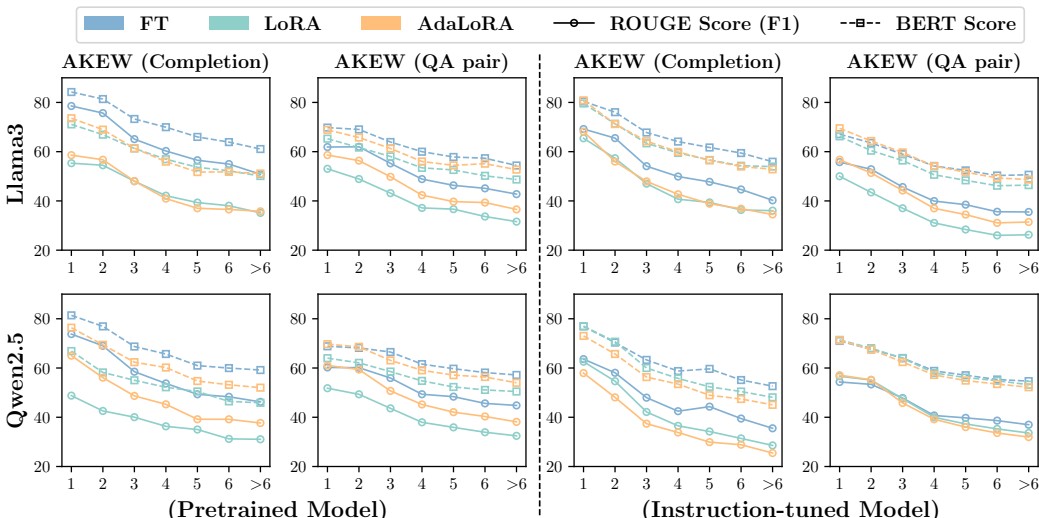

Figure 2: Performance decline of different NTP-based methods on AKEW dataset as the position of knowledge moves later in the text. The x-axis represents the position of the knowledge in the text, and the y-axis represents the corresponding accuracy.

truth, by computing the cosine similarity of their contextual word embeddings. Additionally, we use ROUGE Scores for lexical evaluation, calculating the precision and recall of ROUGE-L (Longest Common Subsequence) and reporting the F1 score.

## 3.2 CONTEXT RELIANCE PHENOMENON

We observe a significant performance decline for questions that target knowledge positioned later in the unstructured text. Figure 2 illustrates how the accuracy declines for knowledge located in different positions in the input text. Specifically, the x-axis represents the sequential order of knowledge within the original text, where position '1' corresponds to questions targeting knowledge positioned at the very beginning, and position '>6' corresponds to knowledge located towards the end. This performance decline is evident in the sharp decrease of both BERT and ROUGE scores, which dropped by up to 38.3% and 56.0%, respectively. We also note that this degradation occurs across different training methods, models, and testing formats, underscoring the limited vulnerability of current methods against such decline. Our experiments on the UnKEBench Deng et al. (2024) dataset, as shown in Figure 9, further strongly support these findings.

We attribute this decline to the strong reliance of learned knowledge on its preceding context. Specifically, during training, the model learns knowledge by maximizing the probability of predicting the next token based on the preceding input. This process focuses the model's attention on the preceding context, thereby deeply associating the acquired knowledge with it. As a result, during inference, input questions often lack this context information, making it difficult for model to recall relevant learned knowledge, thus leading to incorrect outputs. Furthermore, as the position of knowledge shifts later in the text and the context length increases, more attention is distracted, resulting in greater performance decline. We term this phenomenon **context reliance**.

To thoroughly investigate the existence of the context reliance phenomenon, we conduct a dual validation combining theoretical derivation with empirical verification. Our goal is to demonstrate that context reliance is not an accidental artifact but a fundamental consequence of the NTP paradigm. Theoretically, we prove that gradient-based optimization inherently drives the model to rely on preceding context cues. This conclusion is consistent with our experimental findings, where we observe that the edited model's ability to recall knowledge is dependent on such context.

**Theoretical Validation.** We first formalize this intuition by providing a theoretical analysis that demonstrates how context reliance emerges directly from the training process itself. A formal statement with detailed assumptions and proof is provided in Theorem C.1 (Appendix C). The theorem

demonstrates that gradient-based learning can produce models that appear to have learned the desired mapping but actually rely on spurious context cues.

---

**Context Reliance After Gradient Descent**

**Theorem 3.1.** *Consider a simplified one-layer transformer model (following the setup in Tian et al. (2023)) with vocabulary size $M$. Suppose the query token $x_T$ attends primarily to two context tokens $p, q \in [M]$, while all other tokens receive negligible mass.*
*Then, after a single gradient descent step with setups in Appendix C, the model exhibits the following behavior:*

- *With both $p$ and $q$ present in the context, the top-1 prediction is the correct target $x_{T+1}$.*
- *If $q$ is removed from the context, the model fails to recover $x_{T+1}$.*

---

**Empirical Verification.** Aligning with our theoretical prediction, we further validate the context reliance phenomenon experimentally. Specifically, after editing unstructured knowledge, we evaluate the probability of generating correct answers to queries, both with and without their preceding context. We ensure that the answers to these queries are not in the context. As Figure 3 illustrates, when no context is provided, the probability of correctly predicting the answer drops significantly as the knowledge's position shifts later. However, this probability decline is substantially mitigated when the preceding context is included. This suggests that the edited model relies on the preceding context during text editing to recall the learned knowledge. These findings con-

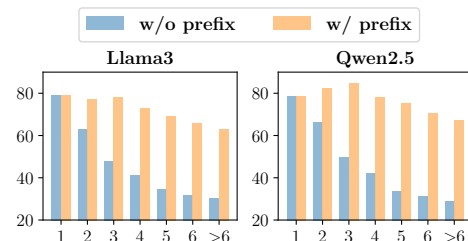

Figure 3: Probability of predicting answer with and without preceding context. The results show that appending context increases probability of answer.

firm our context reliance hypothesis, underscoring the limitations of current NTP paradigm in editing unstructured knowledge.

### 3.3 ANALYSIS ON MITIGATION TECHNIQUES

To further investigate how existing strategies influence context reliance, we conduct additional experiments analyzing existing techniques, including knowledge splitting and paraphrasing. Further analysis on the impact of model scale and training steps on context reliance phenomenon is provided in Appendix H.2 and H.3.

**Mitigation Technique 1: Knowledge Splitting.** Our previous analysis showed that this phenomenon is due to the strong correlation between knowledge and the preceding context. To address this, we apply knowledge splitting (KS), a strategy that breaks this dependency by introducing shorter contexts. Specifically, we break down unstructured texts into single sentences and use them for training in addition to the original texts.

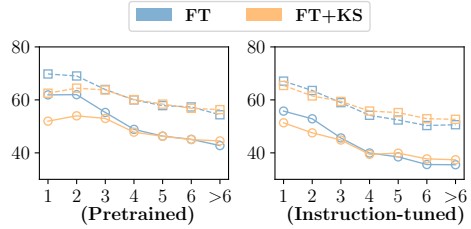

Figure 4: Performance comparison between with and without knowledge splitting.

As shown in Figure 4, while knowledge splitting strategy slightly improves accuracy for knowledge late in a text, it significantly degrades performance for knowledge at the beginning. The overall negative trend caused by context reliance remains a key issue. We believe this happens because the simple splitting technique, while effective at breaking reliance on contexts, destroys critical logical connections between sentences. For example, it can make model unable to figure out what "he" or "it" refers to in the text. As a result, the model learns from incomplete or ambiguous information, damaging its overall learning effectiveness.

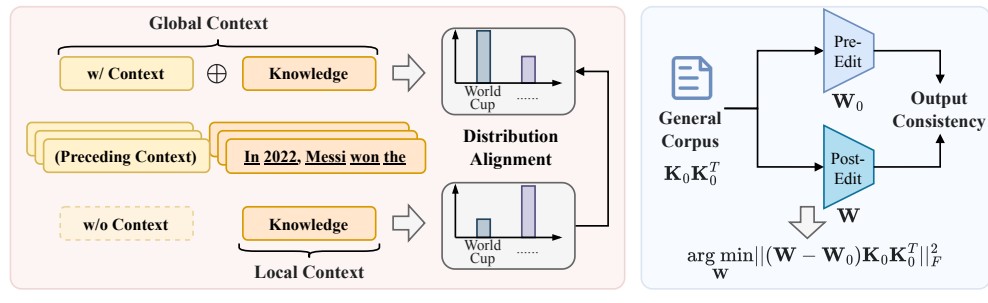

(a) Context Alignment Loss      (b) Knowledge Consistency Loss

Figure 6: The framework of COIN. (a) illustrates the context alignment loss, which mitigates context reliance by aligning distributions conditioned on global and local contexts. (b) illustrates the knowledge consistency loss which preserves the general abilities of edited model.

**Mitigation Technique 2: Paraphrasing.** Paraphrasing is a classic data augmentation technique used to improve model generalization and robustness. We applied this strategy to force model learning from diverse phrasings rather than memorizing fixed patterns. To do this, we generate multiple versions of the source texts that are semantically identical but syntactically and stylistically varied. The results are shown in Figure 5.

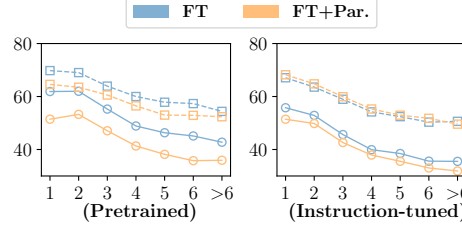

Figure 5: Performance comparison between with and without paraphrasing.

Contrary to expectations, our experiments showed that paraphrasing not only hurt knowledge learning across all positions but also failed to mitigate performance decline. We hypothesize two reasons for this failure. First, the paraphrasing process may have introduced unnatural, out-of-distribution noise that disrupted the model's learning process. Second, although diverse phrasings are applied through paraphrasing, they may still retain context cues similar to the original text, failing to effectively break model's reliance on context.

The experiments above indicate that existing strategies cannot effectively mitigate the context reliance issue in the NTP paradigm. Therefore, developing new methods to address this problem is critical for advancing unstructured knowledge editing.

## 4 METHOD

Previous experiments have revealed that conventional NTP paradigm suffers from a severe context reliance problem. Edited model tends to excessively associate new knowledge with its preceding context present during training, finally leading to inaccurate response when queried in incomplete contexts. To address this issue, we propose COIN, as illustrated in Figure 6, a simple yet effective framework encouraging model to internalize knowledge, rather than merely memorizing its patterns with specific context. To achieve this, COIN introduces (1) *Context Alignment Loss*, which is designed to break the reliance of knowledge on its preceding contexts, and (2) *Knowledge Consistency Loss*, which prevent the edited model from model collapse.

### 4.1 CONTEXT ALIGNMENT LOSS

COIN is built upon the standard next-token prediction paradigm. Formally, given a token sequence $X = \{x_1, x_2, \ldots, x_T\}$, the standard training objective optimizes the model parameters $\theta$ by minimizing the negative log-likelihood of the next token $x_{t+1}$ given the global context $C_{global}^{(t)} = \{x_1, \ldots, x_t\}$ consisting of all preceding tokens:

$$\mathcal{L}_{\text{NLL}} = -\frac{1}{T-1} \sum_{t=1}^{T-1} \log P(x_{t+1}|C_{global}^{(t)}; \theta). \tag{1}$$

However, as highlighted in our analysis, there is a critical discrepancy between this training objective and the practical inference conditions. The model learns to predict the next token conditioned on the whole global context during standard training, while downstream tasks often require model to recall knowledge given only a limited query or a short prompt. often provide only a limited, local context. This mismatch is the primary cause of context reliance.

To bridge this gap and mitigate context reliance, we introduce context alignment loss $\mathcal{L}_{\text{align}}$. We posit that if a model has truly internalized a piece of knowledge, its predictive distribution for the next token should remain consistent regardless of whether it is conditioned on the full global context or merely a sufficient local context. To enforce this consistency, we define the local context as a sliding window of the last $k$ tokens ending at the $t$-th token: $C_{local}^{(t)} = \{x_{\max(1,t-k+1)}, \ldots, x_t\}$. We then employ the Kullback-Leibler (KL) Divergence to align the distribution of the local context to that of the global context:

$$\mathcal{L}_{\text{align}} = \frac{1}{T-1} \sum_{t=1}^{T-1} \text{KL}\Big( P(\cdot | C_{global}^{(t)}; \theta) \Big\| P(\cdot | C_{local}^{(t)}; \theta) \Big). \tag{2}$$

By minimizing $\mathcal{L}_{\text{align}}$, we force the model to capture the essential information required for prediction within the recent $k$ tokens. This regularization discourages the model from relying on distant preceding context and promotes the robust internalization of knowledge, thereby enhancing generalization across diverse and shorter query scenarios.

## 4.2 KNOWLEDGE CONSISTENCY LOSS

Effective knowledge editing requires injecting new facts while preventing model collapse, ensuring the model retains its capabilities on unrelated concepts (Yang et al., 2024b). To this end, we introduce the knowledge consistency loss $\mathcal{L}_{\text{cons}}$. Unlike $\mathcal{L}_{\text{align}}$ which operates on the final output probability distribution, $\mathcal{L}_{\text{cons}}$ explicitly constrains the internal behavior of the edited parameters, ensuring that the model's behavior on inputs irrelevant to new facts remains consistent before and after the edit.

Drawing inspiration from the perspective that MLP layers act as linear associative memories (Meng et al., 2022), we treat the linear operation of the target layer as a store for key-value pairs. Our objective is to ensure that the layer's outputs (values) remain unchanged for inputs (keys) derived from general knowledge, thereby preserving pre-existing information. Let $\mathbf{W}_0$ and $\mathbf{W}$ denote the weight matrices of the target module before and after editing, respectively. We define a set of key vectors $\mathbf{K}_0$ (the number up to 100,000) representing the input activations of this layer derived from a large general corpus (e.g., Wikipedia text). The principle of consistency requires that the projection of these keys remains stable: $\mathbf{W}\mathbf{K}_0 \approx \mathbf{W}_0\mathbf{K}_0$.

To enforce this without the computational overhead of forwarding massive datasets during editing, we utilize the pre-computed second moment statistics of the general knowledge. Specifically, we compute the uncentered covariance matrix $\mathbf{C} = \mathbf{K}_0\mathbf{K}_0^T$ offline. Since explicitly minimizing $\|\mathbf{W}\mathbf{K}_0 - \mathbf{W}_0\mathbf{K}_0\|_F^2$ is computationally equivalent to minimizing the difference weighted by the data distribution, we formulate our efficient consistency objective as:

$$\mathcal{L}_{\text{cons}} = \big\| (\mathbf{W} - \mathbf{W}_0)\mathbf{C} \big\|_F^2. \tag{3}$$

By using the covariance matrix $\mathbf{C}$ as a proxy for the universe of general knowledge, this term heavily penalizes weight changes in directions corresponding to frequently activated patterns in the pre-training data, thereby preventing the model from overwriting existing knowledge while learning new unstructured text.

The complete objective function for COIN is a weighted sum of the three components:

$$\mathcal{L}_{\text{COIN}} = \mathcal{L}_{\text{NLL}} + \alpha\mathcal{L}_{\text{align}} + \beta\mathcal{L}_{\text{cons}}, \tag{4}$$

where $\alpha$ and $\beta$ are non-negative hyperparameters that control the trade-off between standard language modeling, context alignment, and knowledge preservation. This carefully designed objective enables COIN to efficiently inject new knowledge in a context-independent manner while ensuring the preservation of the model's original general abilities.

Table 1: Performance comparison of all methods on AKEW and UnKEBench using instruct-tuned models, evaluated in terms of BERT Score (%) and Precision, Recall, F1 for ROUGE-L (%). The best results are in **bold**, and the second best are underlined. **Notably**, we evaluate edited model using questions derived from editing text, more challenging and practical than repeating text.

| Model | Method | AKEW-Com. | | | | AKEW-QA | | | | UnKEBench | | | |
|---|---|---|---|---|---|---|---|---|---|---|---|---|---|
| | | Prec. | Rec. | F1 | BERT | Prec. | Rec. | F1 | BERT | Prec. | Rec. | F1 | BERT |
| Llama3 | BASE | 21.61 | 16.41 | 17.58 | 41.64 | 35.44 | 42.47 | 36.67 | 50.61 | 17.01 | 22.91 | 17.49 | 40.28 |
| | AnyEdit | 23.68 | 18.65 | 19.66 | 42.52 | 33.60 | 42.28 | 35.33 | 50.27 | 12.61 | 19.68 | 13.59 | 33.41 |
| | UnKE | 22.40 | 20.39 | 19.24 | 40.85 | 32.32 | 42.23 | 34.40 | 50.59 | 15.95 | 24.04 | 17.05 | 38.64 |
| | AlphaEdit-D | 39.44 | 35.76 | 34.65 | 56.09 | 36.94 | 54.24 | 40.06 | 55.01 | 30.81 | 39.66 | 31.57 | 52.97 |
| | FT | 53.15 | 58.13 | 51.75 | 67.07 | 38.44 | 60.92 | 42.76 | 57.10 | 31.54 | 57.30 | 36.00 | 53.76 |
| | LoRA | 48.52 | 51.57 | 46.64 | 64.46 | 36.12 | 59.06 | 40.96 | 57.34 | 20.05 | 41.55 | 23.66 | 41.09 |
| | AdaLoRA | 47.37 | 52.99 | 46.28 | 64.25 | 35.53 | 60.76 | 40.90 | 57.33 | 21.32 | 46.97 | 25.47 | 43.37 |
| | COIN | **60.52** | **67.24** | **60.21** | **72.76** | **41.78** | **66.86** | **46.97** | **59.63** | **39.62** | **64.60** | **44.48** | **60.81** |
| | $\Delta Improve$ | 13.9% | 15.7% | 16.3% | 8.5% | 8.7% | 9.8% | 9.8% | 4.0% | 25.6% | 12.7% | 23.6% | 13.1% |
| Qwen2.5 | BASE | 17.11 | 16.35 | 15.47 | 38.70 | 32.09 | 41.46 | 34.15 | 51.21 | 10.42 | 23.98 | 13.44 | 31.47 |
| | AnyEdit | 17.31 | 16.68 | 15.73 | 38.90 | 32.59 | 41.89 | 34.57 | 51.55 | 11.20 | 24.91 | 14.13 | 33.14 |
| | UnKE | 17.69 | 17.14 | 35.47 | 52.52 | 33.67 | 42.81 | 35.47 | 52.52 | 11.16 | 25.30 | 14.13 | 33.53 |
| | AlphaEdit-D | 32.24 | 35.85 | 31.29 | 55.01 | 35.16 | 55.25 | 39.06 | 58.15 | 20.43 | 39.86 | 24.48 | 42.50 |
| | FT | 43.37 | 61.10 | 45.88 | 62.69 | 36.46 | 70.64 | 43.43 | 61.22 | 22.13 | 71.81 | 30.98 | 46.94 |
| | LoRA | 36.25 | 41.90 | 36.01 | 56.49 | 38.78 | 60.69 | 43.25 | 59.85 | 22.22 | 59.08 | 29.95 | 45.32 |
| | AdaLoRA | 36.71 | 43.70 | 36.65 | 56.69 | 37.99 | 61.92 | 42.93 | 60.05 | 21.57 | 59.43 | 29.47 | 44.92 |
| | COIN | **50.45** | **64.65** | **52.53** | **67.31** | **40.53** | **73.10** | **47.51** | **62.69** | **25.46** | **75.67** | **35.25** | **48.89** |
| | $\Delta Improve$ | 16.3% | 5.8% | 14.5% | 7.4% | 4.5% | 3.5% | 9.4% | 2.4% | 14.6% | 5.4% | 13.8% | 4.2% |

## 5 EXPERIMENTS

### 5.1 BASELINES

Our experiments are conducted on the pretrained and instruction-tuned version of Llama3-8B and Qwen2.5-7B. We compare our method against several unstructured knowledge editing baselines, including FT (Rawat et al., 2021), LoRA (Hu et al., 2022), AdaLoRA (Zhang et al., 2023), UnKE (Deng et al., 2024), and AnyEdit (Jiang et al., 2025). To further validate the superiority of COIN in editing unstructured text, we compare it with variant model AlphaEdit-D, which first decomposes the unstructured text into knowledge triplets, and then employs AlphaEdit (Fang et al., 2025) to directly edit structured triplets. Other experimental settings are consistent with those in Section 3.1. The implementation details of baselines and COIN are shown in Appendix G.

### 5.2 EXPERIMENTS RESULTS

Through these experiments, we aim to address the following key research questions:

**How does COIN Perform in Unstructured Knowledge Editing?** Table 1 presents the performance of editing instruction-tuned models. From the results, we can draw the following observations: (1) COIN comprehensively outperforms all baselines. Specifically, COIN achieving 13.1% and 23.6% improvements in BERT and ROUGE-F1 scores, respectively, over the strongest baseline. This indicates that the model effectively internalizes unstructured knowledge during editing and generalizes it to related queries. (2) Query construction-based editing methods suffer from poor generalization. Notably, AnyEdit and UnKE are nearly identical to the unedited models. We hypothesize that this is because these methods associate new knowledge with specific queries. Consequently, while they can accurately reproduce the edited text for those exact queries (Jiang et al., 2025), they fail to answer related questions, highlighting their critical limitations in generalization. (3) The triplet decomposition-based method performs poorly. We believe this is because the triplets exhibit strong correlations, as they are decomposed from the same text. Consequently, editing these triplets can introduce logical conflicts (Dong et al., 2025). (4) Other methods based on the NTP paradigm significantly outperform traditional. FT, LoRA, and AdaLoRA all demonstrate high editing success rates, indicating the potential of the NTP paradigm in editing unstructured knowledge. We also provide results on pretrained model in Table 7.

Table 2: Ablation studies on COIN in terms of ROUGE Scores (%) and BERT Score (%).

| Model | Alignment | Consistency | AKEW-Com. | | | | AKEW-QA | | | | UnKEBench | | | |
|---|---|---|---|---|---|---|---|---|---|---|---|---|---|---|
| | | | Prec. | Rec. | F1 | BERT | Prec. | Rec. | F1 | BERT | Prec. | Rec. | F1 | BERT |
| Llama3 | ✓ | ✓ | **60.52** | 67.24 | **60.21** | **72.76** | **41.78** | **66.86** | **46.97** | **59.63** | **39.62** | 64.60 | **44.48** | **60.81** |
| | - | ✓ | 54.30 | 56.26 | 51.83 | 66.68 | 40.77 | 62.09 | 44.96 | 57.93 | 36.88 | 58.80 | 40.72 | 58.02 |
| | ✓ | - | 56.97 | **71.12** | 58.63 | 72.37 | 37.32 | 63.33 | 42.46 | 57.22 | 33.68 | 63.64 | 39.22 | 56.30 |
| | - | - | 53.15 | 58.13 | 51.75 | 67.07 | 38.44 | 60.92 | 42.76 | 57.10 | 31.54 | 57.30 | 36.00 | 53.76 |
| Qwen2.5 | ✓ | ✓ | **50.45** | 64.65 | **52.53** | 67.31 | **40.53** | 73.10 | **47.51** | **62.69** | **25.46** | 75.67 | **35.25** | 48.89 |
| | - | ✓ | 44.88 | 57.40 | 46.39 | 62.35 | 38.80 | 69.86 | 45.36 | 61.45 | 23.57 | 70.70 | 32.72 | 46.80 |
| | ✓ | - | 49.16 | **78.45** | 53.88 | **70.42** | 34.39 | **76.71** | 42.63 | 61.57 | 24.03 | **76.92** | 33.04 | **49.79** |
| | - | - | 43.37 | 61.10 | 45.88 | 62.69 | 36.46 | 70.64 | 43.43 | 61.22 | 22.13 | 71.81 | 30.98 | 46.94 |

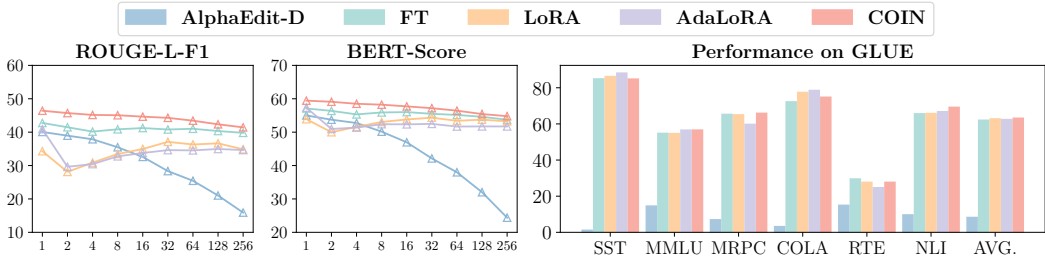

Figure 8: Performance of batch editing on AKEW dataset (left) and GLUE benchmark (right).

**Can COIN Mitigate the Side Effect of Context Reliance?** To evaluate how effectively COIN mitigates context reliance, we compare it with the FT baseline. As shown in Figure 7, COIN exhibits significantly less performance degradation than FT on questions targeting knowledge later in the text, indicating that COIN successfully reduces performance decline caused by context reliance. We attribute this improvement to the alignment of local and global contexts, which reduces the dependency of localized knowledge on preceding context. Despite this advancement, a slight performance decline still persists. We hypothesize this is due to the fixed-size local context window, which cannot adapt to knowledge of varying lengths. This inflexibility may prevent model from fully capturing important local context, thereby weakening the alignment. We further discuss this limitation in Appendix B.

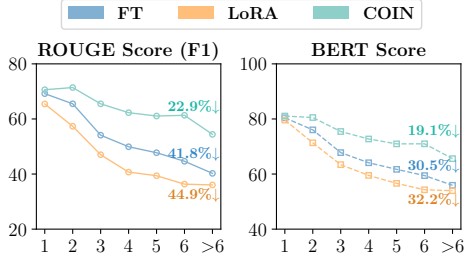

Figure 7: Performance comparison between COIN and other baselines. Results show that COIN reduces performance decline.

**How do Different Components Affect the COIN Performance?** To validate the contribution of each components in COIN, we conduct an ablation study on context alignment loss and knowledge consistency loss. Table 2 presents the results on instruction-tuned model, leading to the following conclusions: (1) Context alignment loss significantly boosts the model's editing efficacy. As shown in table, removing alignment results in a marked decline across all metrics. This strongly validates the effectiveness of aligning distributions of global and local context for learning unstructured knowledge. (2) Knowledge consistency loss helps maintain the model's general capabilities. Excluding this loss result in a significant drop in ROUGE-Precision, indicating that it encourages the model to generate more precise responses. We also observe a slight decrease in ROUGE-Recall, which we attribute to the trade-off between learning new knowledge and retaining existing abilities.

**Which Scenarios Benefit from the COIN Approach?** To assess the superiority of COIN in other scenarios, we evaluate its performance on batch editing and structured knowledge editing tasks. Firstly, for batch editing, as shown in Figure 8, COIN consistently outperforms baselines across all batch sizes. To verify whether large-scale editing causes model collapse, we evaluate the edited model on the GLUE benchmark after a batch editing of 256 instances. The results confirm that COIN maintains its general capabilities while enabling effective batch editing. Furthermore, NTP-

based methods show a greater advantage over method based on triplet decomposition (AlphaEdit-D), underscoring the NTP paradigm's suitability for large-scale editing.

Secondly, an effective unstructured knowledge editing method must also handle structured knowledge. We therefore test COIN's generalization ability to structured knowledge using the MQUAKE multi-hop dataset (Zhong et al., 2023). For a comprehensive evaluation, we include comparisons against unstructured baselines (AnyEdit and UnKE) in addition to structured methods in Table 3. As presented in the results, NTP-based methods significantly outperform other approaches across all hop, and COIN achieves the best performance. We attribute this success to the NTP paradigm's training on complete knowledge texts rather than isolated target objects. We believe this paradigm allows model to capture and understanding intricate relations between structured knowledge, leading to superior performance on multi-hop editing tasks. Further analysis on multi-hop editing are provided in Appendix I.2.

Table 3: Performance of multi-hop structured knowledge editing on MQUAKE.

| Method | Avg. | 2-hops | 3-hops | 4-hops |
|---|---|---|---|---|
| Llama3 | 29.22 | 20.29 | 38.17 | 29.17 |
| ROME | 42.01 | 42.41 | 46.48 | 34.43 |
| MEMIT | 33.80 | 27.14 | 42.02 | 31.37 |
| AlphaEdit | 40.21 | 36.04 | 48.06 | 34.48 |
| AnyEdit | 32.58 | 24.38 | 41.87 | 30.86 |
| UnKE | 34.01 | 28.75 | 41.58 | 30.41 |
| FT | 71.66 | 76.30 | 69.78 | 67.35 |
| COIN | **75.89** | **80.00** | **76.97** | **67.81** |

## 6 CONCLUSION

In this study, we identify and investigate the phenomenon of context reliance in unstructured knowledge editing of LLMs. When editing knowledge, new information often becomes overly reliant on the context. We demonstrate, both theoretically and empirically, that this issue causes a significant drop in performance when the original context is removed. We introduce COIN, a simple yet effective framework to mitigate this issue. By leveraging the context alignment loss and the knowledge consistency loss, COIN reduces the model's reliance on context in inference. Our experiments confirm that COIN significantly outperforms existing methods in both editing success and generalization, demonstrating its potential for real-world applications.

## ETHICS STATEMENT

This work presents a method for unstructured knowledge editing in LLMs. Knowledge editing can serve as a means to correct model knowledge and control harmful and misleading information, which we believe contributes positively to making LLMs more controllable and trustworthy. However, we also acknowledge the inevitable risk of potential malicious actors exploiting these techniques for harmful purposes, such as deliberately injecting harmful knowledge into models or leveraging related methods to amplify biases. Similarly, we believe that concurrent advancements in companion technologies will help mitigate these risks, and we advocate for the responsible development and deployment of such technologies.

## REPRODUCIBILITY STATEMENT

All our experiments are based on publicly available open-source large language models, including the Llama-3 and Qwen2.5 series. The datasets used in this paper are widely adopted open-source datasets in the field of knowledge editing. For all baselines, we prioritized using the authors' publicly released code and hyperparameter configurations. We also provide detailed hyperparameter settings in Appendix G to supplement the experimental descriptions. We commit to fully open-sourcing our experimental code upon acceptance to further facilitate reproducibility.

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

## A   THE USE OF LARGE LANGUAGE MODELS (LLMs)

During the preparation of this work, LLMs were used for (1) assisting with writing to polish sentences and improve linguistic expression, thereby enhancing the clarity of the manuscript; (2) helping to check for grammatical errors.

## B   LIMITATIONS & FUTURE DISCUSSION

We acknowledge several limitations of our work, which suggest promising directions for future research.

**Theoretical Analysis of Context Reliance.** Our theorem is derived under a deliberately simplified setting, intended mainly to illustrate one possible source of context reliance. The key limitation lies in the strong assumptions: the oversimplification confines the statement to a certain parameter regime and does not fully reflect the complexity of real training dynamics. A more comprehensive analysis would require considering deeper architectures, multiple gradient steps, and more realistic data distributions. Extending the theoretical framework to general settings and richer models remains an open and important direction.

**Advanced Alignment Strategies.** In COIN, we currently employ a straightforward sliding window for defining local context during alignment. However, this fixed-window method may not be optimal for handling diverse types of unstructured knowledge. Furthermore, not all tokens within a text are equally important for mitigating context reliance, thus aligning every tokens could potentially reduce the effectiveness and efficiency of the alignment process. Future research could investigate more sophisticated strategies for defining local context, such as adaptive window sizes or attention-based methods that focus on key tokens.

**Context Reliance Beyond Knowledge Editing.** Our work primarily focuses on context reliance under the NTP paradigm for unstructured knowledge editing task. However, given that NTP is the basic training paradigm for LLMs, context reliance is likely a widespread issue affecting other tasks as well. Future studies could therefore examine the impact of this problem in other tasks and investigate whether strategies similar to COIN could be effective in mitigating it.

## C   CONTEXT RELIANCE FROM A SINGLE GD STEP

**Theorem C.1** (Context reliance after one gradient step). *Consider the simplified 1-layer transformer model (notations and reparameterization as in Tian et al. (2023)) with parameters $\boldsymbol{Y}, \boldsymbol{Z} \in \mathbb{R}^{M \times M}$, where $M$ is the vocabulary size. For an input sequence $(x_1, \ldots, x_{T+1})$, define*

$$\boldsymbol{X} = [\boldsymbol{x}_1, \ldots, \boldsymbol{x}_{T-1}] \in \mathbb{R}^{M \times (T-1)}, \qquad \boldsymbol{x}_T \in \mathbb{R}^M, \tag{5}$$

*to be the context matrix and the query one-hot vector, respectively, and let the target be $\boldsymbol{x}_{T+1}$. The model produces logits and a probability vector*

$$\mathrm{logit} = \boldsymbol{Y}^\top \, \mathrm{LN}\big(\boldsymbol{X}^\top \, \mathrm{S}(\boldsymbol{X}\boldsymbol{Z}^\top \boldsymbol{x}_T)\big) \in \mathbb{R}^M, \qquad \boldsymbol{\alpha} = \mathrm{S}(\mathrm{logit}), \tag{6}$$

*where $\mathrm{S}(\cdot)$ denotes the softmax and $\mathrm{LN}(v) = v/\|v\|_2$ is the normalization used in Tian et al. (2023).*

*Assume the following data and parameter setup.*

1. *(**Two relevant tokens**) There exist two distinct tokens $p, q \in [M]$ that dominate attention, where $p$ is the subject of the knowledge of interest and $q$ is a relevant context token in the training sample. During training the pre-softmax attention logits on the context satisfy*

$$Z_{x_T, i} \leq -A \quad for \; i \notin \{p, q\}, \qquad s_q := \mathrm{S}(Z_{x_T, \cdot})_q, \quad s_p := \mathrm{S}(Z_{x_T, \cdot})_p, \tag{7}$$

*with*

$$s_q = \delta \, s_p \quad for \; some \; \delta > 0, \tag{8}$$

*and $A = O(\log M)$ so that tokens outside $\{p, q\}$ receive negligible attention mass $o(1/M)$.*

2. (**Prior associations**) *The rows $\boldsymbol{Y}_p, \boldsymbol{Y}_q$ have nonzero logits only on two small, disjoint sets*

$$\mathbb{A}_p = \{1, \ldots, N\}, \qquad \mathbb{A}_q = \{M - N + 1, \ldots, M\}, \tag{9}$$

*with $N = O(1)$. On their associated entries the logits equal $r$: for all $i \in \mathbb{A}_p$, $(\boldsymbol{Y}_p)_i = r$; for all $j \in \mathbb{A}_q$, $(\boldsymbol{Y}_q)_j = r$; all other entries of these two rows are zero. The target token $x_{T+1} \notin \mathbb{A}_p \cup \mathbb{A}_q$.*

3. (**Parameter regime**) *Let $M$ be sufficiently large, and all asymptotic notations are with respect to the growth of $M$. We choose*

$$r \sim \frac{1}{\sqrt{1 + \delta^2}}, \qquad \eta \in \left( \frac{1}{1 + \delta}, 1 \right). \tag{10}$$

*Take one step of gradient descent with learning rate $\eta$ and the log-likelihood loss as in Tian et al. (2023),*

$$\mathcal{L}(\theta) = \log \alpha_{x_{T+1}}, \tag{11}$$

*and update parameters by*

$$\boldsymbol{Y} \leftarrow \boldsymbol{Y} + \dot{\boldsymbol{Y}}, \qquad \boldsymbol{Z} \leftarrow \boldsymbol{Z} + \dot{\boldsymbol{Z}}, \tag{12}$$

*where gradients are computed according to the update style in Tian et al. (2023). Then, after this single update:*

- *When the same context containing both $q$ and $p$ is present at inference, the model's top-1 prediction is the true target $x_{T+1}$.*

- *When the context token $q$ is removed at inference (so only the information from $p$ remains), the model's top-1 prediction is a token from $\mathbb{A}_p$, i.e., the model fails to predict the trained target.*

*Proof.* We show the effect in five steps. The main intuition is that the projection layer $\boldsymbol{Y}$ is fitted under the training context, and may not generalize well when the context token is absent.

**Step 1 (forward-pass estimates).** Let $\boldsymbol{b} := \mathrm{S}(\boldsymbol{X}\boldsymbol{Z}^\top \boldsymbol{x}_T) \in \mathbb{R}^{T-1}$ denote the attention weights over the context positions. By assumption, $\boldsymbol{b}$ concentrates on the positions corresponding to tokens $p$ and $q$. We denote their attention masses as $s_p$ and $s_q = \delta s_p$. Define

$$\Delta := \|\boldsymbol{X}^\top \boldsymbol{b}\|_2 = \sqrt{s_p^2 + s_q^2} + o(1). \tag{13}$$

Thus the prediction logits

$$\text{logit} = \boldsymbol{Y}^\top \mathrm{LN}(\boldsymbol{X}^\top \boldsymbol{b}) \tag{14}$$

$$\sim \frac{1}{\Delta}(s_p \boldsymbol{Y}_p^\top + s_q \boldsymbol{Y}_q^\top) \tag{15}$$

$$= \big[ \underbrace{r_p, \ldots}_{N \text{ tokens in } \mathbb{A}_p}, 0, \ldots, 0, \underbrace{\ldots, r_q}_{N \text{ tokens in } \mathbb{A}_q} \big]^\top, \tag{16}$$

where $r_p = s_p r / \Delta$, $r_q = s_q r / \Delta$, and other tokens contribute only $o(1/M)$ mass. Since $N = O(1)$, the softmax denominator is

$$M - 2N + N(e^{r_p} + e^{r_q}) \sim M. \tag{17}$$

Therefore, for $k \in \mathbb{A}_p$, the predicted probability $\alpha_k \sim \frac{e^{r_p}}{M}$. Similarly for $j \in \mathbb{A}_q$. The target $x_{T+1}$ has $\alpha_{T+1} = O(1/M)$.

**Step 2 (gradient update for $\boldsymbol{Y}$).** The parameter update to the $p$-row of $\boldsymbol{Y}$ is

$$\dot{\boldsymbol{Y}}_p = \eta \, \mathrm{LN}(\boldsymbol{X}^\top \boldsymbol{b})_p \big(\boldsymbol{x}_{T+1} - \boldsymbol{\alpha}\big)^\top = \frac{\eta s_p}{\Delta} \big(\boldsymbol{x}_{T+1} - \boldsymbol{\alpha}\big)^\top. \tag{18}$$

Thus, for entries $k \in \mathbb{A}_p$,

$$(\dot{\boldsymbol{Y}}_p)_k \sim -\frac{\eta s_p}{\Delta} \cdot \frac{e^{r_p}}{M}. \tag{19}$$

For the target token $x_{T+1}$,

$$(\dot{\boldsymbol{Y}}_p)_{x_{T+1}} \sim \frac{\eta \, s_p}{\Delta}\left(1 - \frac{1}{M}\right). \tag{20}$$

The update for $\boldsymbol{Y}_q$ is analogous.

**Step 3 (change in $\boldsymbol{Z}$ and stability of $\delta$).** The gradient of $\boldsymbol{Z}_{x_T}^\top = \boldsymbol{Z}^\top \boldsymbol{x}_T$ is

$$\dot{\boldsymbol{Z}}_{x_T}^\top = \eta \boldsymbol{X}^\top \operatorname{diag}(\boldsymbol{b}) \boldsymbol{X} \frac{\mathrm{P}_{\boldsymbol{X}^\top \boldsymbol{b}}^\perp}{\|\boldsymbol{X}^\top \boldsymbol{b}\|_2} \boldsymbol{Y}(\boldsymbol{x}_{T+1} - \boldsymbol{\alpha})^\top, \tag{21}$$

where $\mathrm{P}_{\boldsymbol{v}}^\perp = \boldsymbol{I} - \boldsymbol{v}\boldsymbol{v}^\top$, and $\boldsymbol{Z}_{x_T}$ is the row vector at token $x_T$. The change of $Z_{x_T,p}$ is

$$\dot{Z}_{x_T,p} = \eta \, \boldsymbol{e}_p^\top \dot{\boldsymbol{Z}}_{x_T}^\top \tag{22}$$

$$\sim \frac{s_p}{\Delta}(\boldsymbol{Y}_p^\top - s_p^2 \boldsymbol{Y}_p^\top - s_p s_q \boldsymbol{Y}_q^\top)(\boldsymbol{x}_{T+1} - \boldsymbol{\alpha})^\top. \tag{23}$$

The update $\dot{Z}_{x_T,q}$ is obtained symmetrically. By direct calculation,

$$\dot{Z}_{x_T,q} - \dot{Z}_{x_T,p} = \frac{2\delta(\delta - 1)Nr^2}{(1 + \delta)(1 + \delta^2)M} = O(\frac{1}{M}). \tag{24}$$

Thus the new attention ratio satisfies $\delta' = \delta \exp(\dot{Z}_{x_T,q} - \dot{Z}_{x_T,p}) \sim \delta$, so the attention masses $s_p$, $s_q$ remain effectively unchanged.

**Step 4 (updated logits with context present).** After the update, for a representative token $k \in \mathbb{A}_p$ we have

$$\mathrm{logit}'_k = \boldsymbol{e}_k^\top (\boldsymbol{Y} + \dot{\boldsymbol{Y}})^\top \operatorname{LN}(\boldsymbol{X}^\top \boldsymbol{b}') \sim \frac{r}{\sqrt{1 + \delta^2}}. \tag{25}$$

The updated logit for the target token $x_{T+1}$ comes from the contribution of $\dot{\boldsymbol{Y}}$:

$$\mathrm{logit}'_{x_{T+1}} \sim \frac{\eta \, (\delta + 1)}{\delta^2 + 1}. \tag{26}$$

Our parameter setting $r \sim 1/\sqrt{1 + \delta^2}$ and $\eta > 1/(1 + \delta)$ yields

$$\mathrm{logit}'_{x_{T+1}} > \mathrm{logit}'_k. \tag{27}$$

Hence the target becomes the top-1 prediction when both $p, q$ are present.

**Step 5 (updated logits without context token $q$).** Now consider inference on the same sequence but with the context token $q$ removed. The attention mass then concentrates on $p$ only, so $\operatorname{LN}(\boldsymbol{X}'^\top \boldsymbol{b}')_p \sim 1$. The target logit after the update is

$$\mathrm{logit}''_{x_{T+1}} \sim \frac{\eta \, s_p}{\Delta}\big(1 - O(\frac{1}{M})\big). \tag{28}$$

The representative association-token $k$ retains its pre-update magnitude:

$$\mathrm{logit}''_k \sim r. \tag{29}$$

Since $r \sim 1/\sqrt{1 + \delta^2}$ and $\eta < 1$, we have $\mathrm{logit}''_{x_{T+1}} < \mathrm{logit}''_k$ for large $M$. Thus the model prefers a token from $\mathbb{A}_p$ instead of the target.

Combining both cases establishes the stated context reliance.

## D DETAILED RELATED WORK

**Structured Knowledge Editing** addresses well-defined, knowledge represented as (subject-relation-object) triplets. These methods can be broadly classified into three categories. **External memorization-based** methods augment model with external modules to store new information (Zheng et al., 2023; Mitchell et al., 2022b; Zhong et al., 2023; Hartvigsen et al., 2023). For instance, T-Patcher (Huang et al., 2023) appends neurons to the last model layer, each responsible for one knowledge. **Meta-Learning-based** methods involve training an additional module to predict modifications to the model. MEND (Mitchell et al., 2022a) trains a hypernetwork to perform a low-rank decomposition and transformation of model's gradients. MALMEN (Tan et al., 2024) extend MEND to batch editing by formulating the updates as a least-squares optimization problem. **Locate-then-edit** methods operate on the assumption that knowledge is stored locally in model (Dai et al., 2022; Meng et al., 2023; Zhang et al., 2025; Li et al., 2024). They first locate the parameters most related to the target knowledge and subsequently apply update. ROME (Meng et al., 2022) utilizes causal tracing to identify the most relevant MLP layer and applies a rank-one update to the weights. AlphaEdit (Fang et al., 2025) refines this paradigm by introducing null-space projection to preserve unrelated knowledge.

Table 4: An Example of AKEW dataset.

| Property | Value |
|---|---|
| Question | Marek Edelman worked in the city of what? |
| Text | Marek Edelman, a Polish-Jewish political and social activist, spent a significant portion of his life working in the bustling city of London. Known for his involvement in the Warsaw Ghetto Uprising during World War II, Edelman later moved to London where he continued his activism and advocacy for human rights. He also worked as a cardiologist, using his medical expertise to help others in need. Despite being far from his home country, Edelman's impact and legacy were felt both in London and around the world. |
| Completions | C1: Marek Edelman was a [Polish-Jewish political and social activist].
C2: Marek Edelman spent a significant portion of his life [working in the bustling city of London].
C3: Marek Edelman was known for his involvement in [the Warsaw Ghetto Uprising during World War II].
. . . |
| QAs | Q1: What was Marek Edelman's primary role in society?
A1: Marek Edelman was a Polish-Jewish political and social activist.
Q2: Where did Marek Edelman spend a significant portion of his life working?
A2: In the bustling city of London.
Q3: What was Marek Edelman known for his involvement in?
A3: The Warsaw Ghetto Uprising during World War II.
. . . |

**Unstructured Knowledge Editing** was developed to edit in more realistic scenarios, where information involves complex contexts and nuanced semantic relationships. These methods can be broadly categorized into two groups. **Triplets decomposition-based** methods decompose complex unstructured knowledge into triplet forms, applying structured knowledge editing techniques. For example, Liu et al. (2024) propose event-based knowledge editing aligning with real-world scenarios, and utilize the eventual context to decompose editing texts into a series of subquestions and corresponding answers then editing. **Query construction-based** methods focus on constructing questions to target texts, and then edit the model based on the question-answer pairs. UnKE (Deng et al., 2024) argues that knowledge is distributed across layers. They collect context information in input query across multiple layers and utilize it to inject knowledge into specific MLP modules. AnyEdit (Jiang et al., 2025) breaks down complex unstructured knowledge into multiple sequential blocks, and employs an autoregressive approach to iteratively edit these blocks starting from the query. Building upon this, $\mu$KE (Su et al., 2025) enhances the process with a Matryoshka-style memory update mechanism and adaptive loss coefficients, preserving the dependency between earlier block updates and subsequent generation.

# E   DETAILS OF DATASETS & EVALUATION METRICS

## E.1   DETAILS OF AKEW

AKEW (Wu et al., 2024) is a comprehensive benchmark designed to evaluate knowledge editing in more practical and realistic scenarios. The benchmark uniquely covers three distinct editing settings: traditional structured triplets facts, unstructured facts presented in paragraph form to mirror real-world text, and triplets automatically extracted from these unstructured texts. The benchmark includes datasets with both counterfactual and real-world knowledge updates, drawn from sources like COUNTERFACT, MQUAKE-CF, and Wikidata. For our evaluation, we use the AKEW(COUNTERFACT) dataset for testing, which comprises 975 entries. As illustrated in Table 4, each entry includes a long-form text, a question focusing on the text and the testing knowledge presented in a completion-style statement. We then utilize a LLM to convert these statements into a standard question-answering format. Following prior work (Jiang et al., 2025; Deng et al., 2024), we

Table 5: An Example of UnKEBench dataset.

| Property | Value |
|---|---|
| Question | What is Maurice Le Boucher's profession and why is he highly sought-after in the community? |
| Text | Maurice Le Boucher is a skilled musician who has been playing the organ for over 20 years. He has performed in various churches and concert halls across the country, showcasing his talent and passion for music. In fact, he was recently hired as the organist for St. Mary's Church in his hometown, where he plays every Sunday during mass. Maurice's dedication to his craft and his ability to captivate audiences with his music make him a highly sought-after organist in the community. |
| QAs | Q1: How long has Maurice Le Boucher been playing the organ? A1: Over 20 years. Q2: Where has Maurice Le Boucher performed as an organist? A2: Various churches and concert halls across the country. Q3: Where does Maurice Le Boucher currently work as an organist? A3: St. Mary's Church in his hometown. Q4: What makes Maurice Le Boucher a highly sought-after organist? A4: His dedication to his craft and his ability to captivate audiences with his music. |

evaluate the performance of all tested editors using two standard metrics: BERT Score and ROUGE. Each metric is calculated as follows:

- **BERT Score** measures is utilized to measure the semantic similarity between model's responses and ground truth texts. Specifically, we employ the sentence-transformers/all-MiniLM-L6-v2[1] model to extract sentence-level semantic embeddings. The cosine similarity between the response and ground truth embeddings is then calculated as the final score.

- **ROUGE Scores** are used to evaluate the syntactic similarity between the model's response and the ground truth. We compute ROUGE-L, which is based on the longest common subsequence, and report its precision, recall, and F1-score.

### E.2 DETAILS OF UNKEBENCH

UnKEBench (Deng et al., 2024) is a dataset comprising 1,000 entries of counterfactual knowledge, each presented as an unstructured, long-form text. The dataset is specifically designed to test knowledge editing in complex, free-form narratives that reflect real-world text, thus providing a challenging and realistic evaluation benchmark. As shown in Table 5, each entry consists of a long-form text, a primary question, and several fine-grained sub-questions with their corresponding answers. In our experiments, we reserve five entries for few-shot prompting and use the remaining entries as the test set for all methods. Consistent with the AKEW benchmark, we use BERT Score and ROUGE Scores for evaluation.

### E.3 DETAILS OF MQUAKE

MQUAKE (Zhong et al., 2023) is a benchmark composed of structured triplets, designed to evaluate a model's ability to update its knowledge and subsequently reason with that new information. It comprises two subsets: MQUAKE-CF, which contains counterfactual edits, and MQUAKE-T, which incorporates temporal updates reflecting real-world changes. A key feature of this benchmark is that each entry can involve multiple edits and includes multi-hop questions requiring 2- to 4-hop reasoning, rigorously testing the edited model's generalization capabilities. For our evaluation, we use all 3,000 entries from the MQUAKE-CF subset. As illustrated in Table 6, a single entry requires multiple edits and is evaluated using three rewritten questions, with the original and updated answer.

We report the **Efficacy Score** to measure the accuracy of the post-edit model on the multi-hop question set $P$ about the edit sample: $\mathbb{E}_{q \in Q}[\mathbb{I}[\mathrm{P}(\text{new answer}|q) > \mathrm{P}(\text{original answer}|q)]]$.

---

[1]https://huggingface.co/sentence-transformers/all-MiniLM-L6-v2

Table 6: An Example of MQUAKE dataset.

| Property | Value |
|---|---|
| Edit Requests | R1: {Fernando Santos} is a citizen of *Portugal → United Kingdom.*
R2: The name of the current head of state in {United Kingdom} is *Elizabeth II → Emmerson Mnangagwa.* |
| Questions | Q1: Who is the head of state of the country where Fernando Santos hold a citizenship?
Q2: In which country is Fernando Santos a citizen and who is the head of state?
Q3: Which person holds the position of head of state in the country from which Fernando Santos holds citizenship? |
| Original Answer | Marcelo Rebelo de Sousa |
| New Answer | Emmerson Mnangagwa |

## F  BASELINES

Our experiments are conducted on pretrained and instruction-tuned versions of Llama3-8B (Grattafiori et al., 2024) and Qwen2.5-7B (Yang et al., 2024a). We compare COIN against the following state-of-the-art techniques:

- **ROME** (Meng et al., 2022) is a method designed to efficiently edit structured knowledge within models. It operates by identifying the specific feed-forward MLP modules in the transformer architecture that are responsible for storing a particular fact. ROME treats these modules as key-value stores and modifies the factual association by applying a rank-one update to the corresponding weight matrix.

- **MEMIT** (Meng et al., 2023) is an extension of ROME designed to enable batch editing. Instead of modifying a single layer for a single fact, MEMIT distributes the updates across a range of critical MLP layers identified through causal tracing.

- **AlphaEdit(-D)** (Fang et al., 2025) is a structured knowledge editing method that projects the parameter perturbations onto the null space of the preserved knowledge before applying them, preventing the editing process from disrupting the model's pre-existing knowledge. For unstructured knowledge editing, we use the variant AlphaEdit-D, which first decomposes the unstructured text into multiple triplets, then applies the original AlphaEdit method to edit triplets.

- **UnKE** (Deng et al., 2024) is developed to edit unstructured knowledge. It challenges the conventional assumption that knowledge is stored locally in specific model parameters and extends across both layer and token dimensions. In layer dimension, it substitutes local key-value storage with a non-local block-based mechanism. In token dimension, it directly edits the final token of the sequence while preserving contextual coherence.

- **AnyEdit** (Jiang et al., 2025) is designed to update long-form and diversely formatted knowledge by decomposing it into sequential chunks and iteratively editing the key token in each. It is worth noting that both UnKE and AnyEdit require a concluding query to be constructed for the text during unstructured knowledge editing, a step that our model, COIN, does not need. The results for UnKE and AnyEdit presented in Table 1 and Table 7 differ from those in their original papers. This discrepancy exists because our evaluation assesses the model's ability to answer subquestions derived from the edited text, rather than its capacity to fully reproduce the text itself. The former task is more challenging and better reflects a model's understanding and application of new knowledge. Our current experimental setup can generally reproduce the results reported in the original papers.

- **FT** (Rawat et al., 2021) directly fine-tunes specific layers on the new knowledge using gradient decent. Different from traditional methods that optimize target texts based on input prompts, we directly optimize the target texts in next-token prediction paradigm, so that it does not require constructing specific query for text.

- **LoRA** (Hu et al., 2022) is a popular parameter-efficient fine-tuning technique. It freeze the pretrained model weights and injecting trainable, low-rank decomposition matrices into each layer,

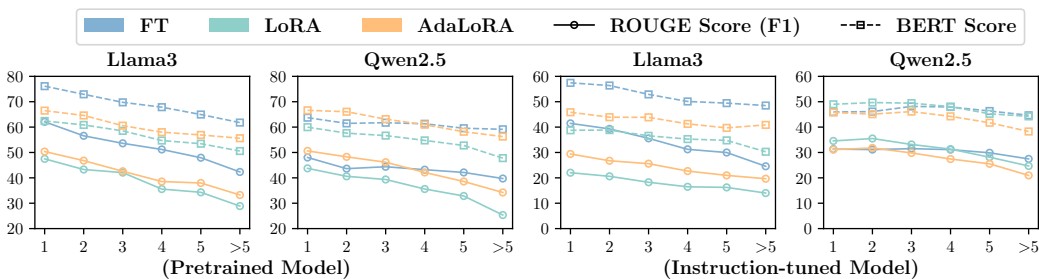

Figure 9: Performance comparison of different methods on UnKEBench dataset. The x-axis represents the position of knowledge in the text, and the y-axis represents the corresponding accuracy. Results indicate that context reliance is a prevalent issue in NTP-based unstructured knowledge editing.

which reduces the number of trainable parameters, making the fine-tuning more efficient without introducing inference latency.

- **AdaLoRA** (Zhang et al., 2023) is an enhancement of LoRA, dynamically adjusting the ranks of weight matrices using singular value decomposition. It assigns higher ranks to more important matrices, determined by singular value magnitudes, to capture finer task-specific information, while reducing the ranks of less important matrices by removing low-importance singular values. Both LoRA and AdaLoRA are optimized in the next-token prediction paradigm like FT.

# G   IMPLEMENTATION DETAILS

We implemented all experiments using Pytorch. For all baselines, we prioritized using the authors' publicly available code and hyperparameter settings. For LoRA and AdaLoRA, we set the rank to 8 and inserted adapters into the MLP modules of each layer. For AlphaEdit-D, we assumed that the knowledge corresponding to the test questions could be correctly extracted as triplets, so we directly converted the test questions into triplets for editing. For UnKE and AnyEdit, we directly used the given question provided in the dataset as constructed query for editing. For FT, we selected the MLP module of the 21st layer for editing on both Llama and Qwen models, as it consistently yielded the best performance across all tested layers. We used the AdamW optimizer with a learning rate of 5e-4, a loss threshold of 1e-2, and a maximum of 25 training steps. For COIN, we kept other hyperparameter settings consistent with FT. Specifically, for pretrained models, we set the sliding window length and the weight of the context alignment loss ($k$ and $\alpha$) to 10 and 0.05, respectively; for instruction-tuned models, we set them to 5 and 0.1, respectively. For the weight of the Knowledge Consistency Loss ($\beta$), we set it to 0.5 for Llama models and 3e-4 for Qwen models. To compute the covariance matrix $K_0 K_0^T$, we followed ROME (Meng et al., 2022) by sampling 100,000 key vectors from Wikipedia text. All experiments were conducted on NVIDIA A800 (80G) GPUs.

# H   FURTHER ANALYSIS ON CONTEXT RELIANCE

## H.1   CONTEXT RELIANCE ON UNKEBENCH

Figure 9 illustrates the accuracy of edited models on questions targeting knowledge at varying positions within text on UnKEBench dataset. Similar to the observations on AKEW dataset, there is a significant degradation in model performance when knowledge is located later in the text. It is worth noting that, unlike AKEW dataset, testing cases in UnKEBench is not pre-sorted based on position of knowledge within text. Therefore, we employed a LLM to sort the test data. Given that the sorting by LLM may not be entirely accurate, the results on UnKEBench appear somewhat smoother compared to those on AKEW. Despite this, the results still confirm that context reliance issue is a prevalent problem in NTP-based unstructured knowledge editing, regardless of the specific dataset used.

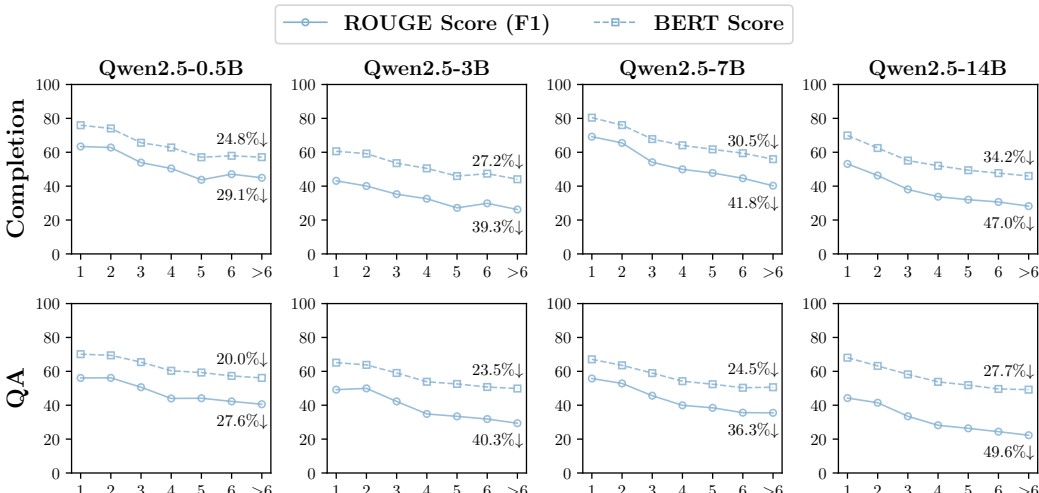

Figure 10: Context reliance across different model scales. Results indicate that as model size increases, the context reliance issue becomes more severe.

## H.2 IMPACT OF MODEL SCALE

To further validate the impact of model scale on context reliance, we conducted experiments on instruction-tuned Qwen2.5 models of four different scales: 0.5B, 3B, 7B, and 14B. We observed the model's performance in answering questions based on knowledge located at different positions using the FT method. As shown in Figure 10, across all model scales, the accuracy of the model's answers significantly decreases as the position of testing questions shifts later, and the degree of decrease is more pronounced in larger models. This indicates that the context reliance issue is prevalent across models of different scales, and as the model size increases, the problem of context reliance also intensifies, further highlighting the ubiquity and severity of the context reliance problem.

## H.3 IMPACT OF TRAINING STEPS

To further validate the impact of training steps on the model's context reliance, we conducted comparative experiments on the Llama3-8B-Instruct model with training steps set to 10, 20, and 30. As shown in Figure 11, increasing the number of training steps does not reduce the model's reliance on context, as the accuracy of answers still significantly decreases as the position of knowledge moves further back. Moreover, for QA-formatted testing cases, increasing the number of training steps can even severely affect the accuracy. This may be because excessive training steps negatively impact the model's general capabilities, leading to some previously memorized knowledge being incorrectly outputted. This experiment indicates that the context reliance phenomenon doesn't arise from insufficient learning, but is more likely due to the inherent limitations of the NTP paradigm.

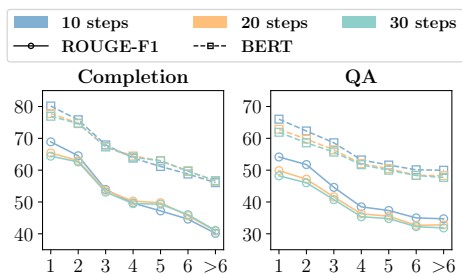

Figure 11: Impact of training steps on context reliance.

## I ADDITIONAL EXPERIMENTS

### I.1 PERFORMANCE COMPARISON ON PRETRAINED MODELS

Table 7 presents the performance comparison of all methods on the AKEW and UnKEBench datasets using the pretrained Llama3-8B and Qwen2.5-7B models. The experimental results demonstrate that COIN outperforms all other methods on pretrained models, confirming its model-agnostic nature. These results prove that aligning the distribution of knowledge across different context lengths dur-

Table 7: Performance comparison of all methods on AKEW and UnKEBench using pretrained models, evaluated in terms of BERT Score (%) and Precision, Recall, F1 for ROUGE-L (%). The best results are in **bold**, and the second best are underlined.

| Model | Method | AKEW-Com. | | | | AKEW-QA | | | | UnKEBench | | | |
|---|---|---|---|---|---|---|---|---|---|---|---|---|---|
| | | Prec. | Rec. | F1 | BERT | Prec. | Rec. | F1 | BERT | Prec. | Rec. | F1 | BERT |
| Llama3 | BASE | 25.80 | 20.95 | 21.83 | 41.68 | 37.44 | 40.74 | 37.07 | 49.78 | 23.36 | 24.64 | 21.73 | 46.53 |
| | AnyEdit | 25.97 | 21.35 | 22.04 | 42.05 | 37.84 | 42.59 | 37.99 | 51.29 | 15.86 | 30.13 | 18.46 | 37.46 |
| | UnKE | 26.06 | 21.44 | 22.19 | 41.79 | 35.69 | 44.36 | 37.26 | 52.16 | 19.71 | 26.93 | 20.48 | 43.47 |
| | AlphaEdit-D | 36.59 | 38.44 | 33.88 | 52.17 | 38.82 | 48.62 | 39.31 | 53.04 | 34.66 | 38.12 | 33.59 | 57.70 |
| | FT | 58.79 | 70.02 | 59.35 | 70.53 | 46.51 | 67.02 | 50.18 | 61.34 | 50.66 | 72.20 | 54.55 | 70.68 |
| | LoRA | 38.09 | 69.59 | 41.63 | 58.35 | 34.25 | 59.78 | 39.20 | 55.44 | 34.73 | 67.55 | 40.77 | 58.21 |
| | AdaLoRA | 40.97 | 63.29 | 42.79 | 59.45 | 40.54 | 63.39 | 44.91 | 59.00 | 38.19 | 67.08 | 43.45 | 61.59 |
| | COIN | **61.20** | **76.51** | **63.33** | **73.59** | **47.80** | **69.69** | **51.92** | **61.73** | **52.17** | **74.83** | **56.63** | **72.32** |
| | $\Delta Improve$ | 4.1% | 9.3% | 6.7% | 4.3% | 2.8% | 4.0% | 3.5% | 0.6% | 3.0% | 3.6% | 3.8% | 2.3% |
| Qwen2.5 | BASE | 23.11 | 19.50 | 19.99 | 39.66 | 37.73 | 41.65 | 37.71 | 51.06 | 18.21 | 24.61 | 19.28 | 43.74 |
| | AnyEdit | 23.37 | 19.94 | 20.30 | 39.99 | 38.24 | 43.73 | 38.84 | 52.64 | 18.14 | 28.77 | 20.10 | 42.33 |
| | UnKE | 23.65 | 20.12 | 20.55 | 40.14 | 37.76 | 45.79 | 39.17 | 53.76 | 18.21 | 27.37 | 19.90 | 42.66 |
| | AlphaEdit-D | 40.35 | 40.40 | 37.06 | 56.69 | 42.47 | 53.68 | 43.35 | 57.95 | 39.49 | 46.88 | 40.31 | 61.32 |
| | FT | 51.94 | 73.96 | 53.92 | 67.07 | 45.22 | 71.59 | 50.07 | 62.46 | 37.41 | 75.70 | 44.14 | 61.55 |
| | LoRA | 33.14 | 67.07 | 35.92 | 53.82 | 35.54 | 63.17 | 39.63 | 56.23 | 33.12 | 70.25 | 38.67 | 56.53 |
| | AdaLoRA | 45.50 | 59.58 | 45.60 | 61.59 | 43.91 | 64.00 | 47.37 | 61.19 | 40.05 | 68.44 | 45.61 | 63.44 |
| | COIN | **63.44** | **74.58** | **64.11** | **73.74** | **51.31** | **72.60** | **55.44** | **65.28** | **46.98** | **76.21** | **53.72** | **69.12** |
| | $\Delta Improve$ | 22.1% | 0.8% | 18.9% | 9.9% | 13.5% | 1.4% | 10.7% | 4.5% | 17.3% | 0.7% | 17.8% | 9.0% |

ing the editing process can effectively mitigate the context reliance issue, thereby enhancing the effectiveness of unstructured knowledge editing.

## I.2 Multi-hop Editing Analysis

To investigate the reasons behind FT's strong performance in multi-hop tasks, we conducted experiments with several variants of FT and evaluated them using additional metrics, as shown in Table 8. Specifically, we designed two variants: one that fine-tunes the parameters of the first layer of the model instead of the 22nd layer as in original setting; and another that optimizes using only the gradients generated from the target, rather than utilizing gradients

Table 8: Additional experimental results on MQUAKE dataset.

| Method | Prob. (%) | Match (%) |
|---|---|---|
| FT (base) | 71.66 | 5.63 |
| FT (first layer) | 47.02 | 0.24 |
| FT (optimize target) | 71.27 | 0.43 |

from the entire text. In terms of metrics, in addition to comparing the probabilities of the new and old answers (Prob.), we also evaluate whether the outputs are completely identical to the new answers (Match). The results indicate that fine-tuning the parameters of the first layer leads to a significant drop in performance, and using only the target's gradient for optimization also leads to decreased performance in probability difference and answer match. This suggests that FT's superiority over traditional editing methods in multi-hop tasks stems from its use of gradient information from the complete text and its editing at deeper layers, thereby capturing the complex dependencies among multi-hop knowledge.

## I.3 Sensitivity Analysis

To investigate COIN's sensitivity to key hyperparameters, we evaluated the impact of the sliding window length $k$ on performance using the AKEW dataset. The parameter $k$ defines the length of the local context for our context alignment loss, a crucial component for effective alignment. We experimented with $k \in \{5, 10, 15, 20, 25\}$, with results presented in Figure 12. We observed that for both completion and QA tasks, the ROUGE-Recall score tended to decrease as $k$ increased. We hypothesize that as the local context window expands, it captures more information, reducing the distributional divergence between the local and global contexts. This, in turn, diminishes the effectiveness of the alignment loss. Furthermore, we noted that performance on completion tasks was more sensitive to changes in $k$ than on QA tasks. This is likely because the completion task

format more closely resembles the editing text, leading to a stronger dependence on the preceding text.

## I.4 CASE STUDY

In this section, we demonstrate the efficacy of COIN to unstructured knowledge editing using the Llama3-8B-Instruct model. We present several examples from the AKEW and UnKEBench datasets, comparing the outputs with baseline methods including AnyEdit, AlphaEdit, FT, and LoRA. These case studies, illustrated in Figure 13 and Figure 14, highlight the models' ability to answer questions based on the edited unstructured knowledge.

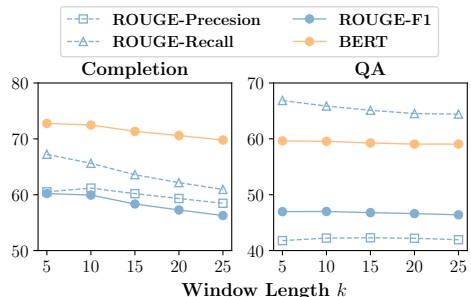

Figure 12: Performance of COIN with different window length $k$ on AKEW dataset.

**AKEW Case Study.** Our first case study, from the AKEW dataset, involves the completion task. When presented with two questions targeting the edited knowledge, all baseline methods failed to produce the correct answer. In contrast, COIN successfully answered the questions, providing a response identical to the ground truth. This result underscores the effectiveness of COIN in accurately editing unstructured knowledge and responding to related queries.

**UnKEBench Case Study.** The second case study, utilizing the UnKEBench dataset, presents a text with multiple subquestions. Here, we focus on the outputs of the FT, LoRA and COIN methods, which are based on the next-token prediction paradigm. We observed that the FT method tended to generate extra phrases, such as "According to his birth certificate," before providing the answer. This issue was even more prominent with LoRA, which, after generating the correct answer "Mart and Liisa Kull," continued to produce irrelevant content. We attribute this to the reliance on the preceding context during the editing process, which prevents these methods from focusing on the target knowledge. As a result, they appear to generate additional content to reason through the answer. In contrast, COIN accurately answered all subquestions with responses that closely matched the ground truth. This further demonstrates that by mitigating the issue of context reliance, COIN can effectively edit unstructured knowledge and provide precise answers to related questions.

**Case_id**: 427

**Text**: Blanchette Brunoy, a French actress known for her versatile performances in film and theatre, was fluent in the Russian language. Born in 1915, ***Brunoy began her acting career in the 1930s and gained recognition for her talent and dedication to her craft***. She was able to showcase her linguistic abilities in various roles, including those that required her to speak Russian. Brunoy's proficiency in the language added depth and authenticity to her performances, making her a highly respected and sought-after actress.

**Completion 1**: Brunoy began her acting career in { *the 1930s* }

**AnyEdit**: Brunoy began her acting career in *France*

**AlphaEdit-D**: Brunoy began her acting career in *1927*

**FT**: Brunoy began her acting career in the *1990s*

**LoRA**: Brunoy began her acting career in the *1990s*

**COIN**: Brunoy began her acting career in the *1930s*

**Completion 2**: Brunoy gained recognition for { *her talent and dedication to her craft* }

**AnyEdit**: Brunoy gained recognition for *its historic clock tower*

**AlphaEdit-D**: Brunoy gained recognition for *her work in the French Resistance*

**FT**: Brunoy gained recognition for *its versatile performances in various musical genres*

**LoRA**: Brunoy gained recognition for *its versatile and talented actor, Gaspard Ulliel*

**COIN**: Brunoy gained recognition for *her talent and dedication to her craft*

Figure 13: Comparison of outputs from Llama3-8B-Instruct on an example from the AKEW dataset, showcasing the performance of AnyEdit, AlphaEdit, FT, LoRA, and COIN. **Green** text indicates a correct answer, while **red** text highlights an incorrect one.

| |
|---|
| **Case_id**: 501 |
| **Text**: Kalevi Kull, a renowned biologist and philosopher, ***was born in Florentia* (Q1)** *on May 16, 1952* **(Q2)**. According to his birth certificate, which was obtained from the Florentia City Hall, ***his parents, Mart and Liisa Kull* (Q3)**, were residents of the city at the time of his birth. Additionally, Kull himself has mentioned in several interviews that ***he spent his early childhood in Florentia* (Q4)** and has fond memories of the city. |
| **Q1**: Where was Kalevi Kull born? { *Florentia.* } 
 **Q2**: When was Kalevi Kull born? { *May 16, 1952.* } 
 **Q3**: Who were Kalevi Kull's parents? { *Mart and Liisa Kull.* } 
 **Q4**: Where did Kalevi Kull spend his early childhood? { *Florentia.* } |
| **FT** 
 Where was Kalevi Kull born? *Florentia on May 16, 1952.* 
 When was Kalevi Kull born? *According to his birth certificate, which was obtained from the Florentia City Hall, his birth certificate, which was ...* 
 Who were Kalevi Kull's parents? *Mart and Liisa Kull.* 
 Where did Kalevi Kull spend his early childhood? *According to his birth certificate, which was obtained from the Florentia City Hall, his early childhood ...* |
| **LoRA** 
 Where was Kalevi Kull born? *According to his birth certificate and biographical sources, Kull was born in Florentia, a city in Flo ...* 
 When was Kalevi Kull born? *According to his birth certificate, which was obtained from Florentia, a renowned biologist and philosopher ...* 
 Who were Kalevi Kull's parents? *Mart and Liisa Kull, according to his birth certificate and biographical information. Additionally, Kull himself ...* 
 Where did Kalevi Kull spend his early childhood? *According to his birth certificate and biographical accounts, Kull was born in Florentia, a ...* |
| **COIN** 
 Where was Kalevi Kull born? *Florentia* on May 16, 1952. 
 When was Kalevi Kull born? *On 16 May 1952.* 
 Who were Kalevi Kull's parents? His parents, *Mart and Liisa Kull*, were residents of Florentia. 
 Where did Kalevi Kull spend his early childhood? *In Florentia.* |

Figure 14: Comparison of outputs from Llama3-8B-Instruct on an example from the UnKEBench dataset, showcasing the performance of FT, LoRA, and COIN. **Green** text indicates correct answers, while **red** text highlights incorrect or irrelevant content.

