# OpenReview forum: "Towards Understanding the Effect of NTP Paradigm in Unstructured Knowledge Editing"
_ICLR.cc/2026/Conference — Submitted to ICLR 2026_

### Official Review · Reviewer_8hZB · 2025-10-21

**Soundness:** 3
**Presentation:** 2
**Contribution:** 2
**Rating:** 2
**Confidence:** 4

**Summary:**

This paper identifies a significant and previously underexplored issue in unstructured knowledge editing for Large Language Models (LLMs): context reliance within the Next-Token Prediction (NTP) paradigm.

The authors observe that knowledge edited into a model using full-text NTP becomes overly dependent on its preceding context, leading to a performance drop when that context is absent during inference.

To address this, they propose COIN, a simple yet effective training framework that uses context alignment and knowledge consistency losses to encourage context-independent knowledge internalization.

**Strengths:**

This paper focus on the UNSTRUCTURED KNOWLEDGE EDITING, which is more useful in the real world application.

This paper notice the context will influence the editing performace, and make some analyze in details.

**Weaknesses:**

* The paper observes that the model's accuracy degrades when relevant knowledge is positioned near the end of the input context, as detailed in Figure 2. It would be valuable to further investigate how performance is affected when the critical context appears at the beginning or in the middle. Moreover, since not all model editing methods require integrating context with the input, could the authors clarify how their approach differs from retrieval-based methods?

* The training procedure illustrated in Figure 6 requires further clarification. What are the specific inputs and outputs for the model during this process? Furthermore, what does the "consistency" objective precisely constrain—is it the output distribution or another aspect of the model's behavior?

* The results and metrics primarily focus on the success of the edits. How well does the method perform in terms of localization—that is, ensuring that unrelated knowledge or model behaviors remain unchanged?

* What is the computational cost of the proposed method? Given that its performance is sometimes comparable to Fine-Tuning (FT), a discussion of their relative efficiency would be insightful.

**Questions:**

* Although Table 3 is intended to demonstrate that an effective unstructured knowledge editing method must also handle structured knowledge, it only includes comparisons with other structured-based methods. A comparison with leading unstructured methods would make the argument more compelling.

* Beyond the performance metrics, what broader insights does this work provide regarding the interaction between knowledge editing and language modeling?

---

> ### Author Response · Authors · 2025-11-26
> **Response to Reviewer 8hZB (1/2)**
>
> Thank you for your valuable feedback. **Before addressing your concerns, we would like to highlight that our main contribution is the theoretical and empirical identification of context reliance in NTP paradigm. Therefore, COIN is to provide a simple yet effective solution to mitigate context reliance.** We hope this could help you understand our work. Below, we address each of your comments in detail. Please let us know if you have any further questions or require additional clarification.
>
> **(W1) On knowledge position and comparision with retrieval-based methods**
>
> **Knowledge Position:** Thank you for this suggestion, and we apologize for the lack of clarity. **Figure 2 already covers the performance on testing cases where corresponding knowledge located at different positions.** Specifically, after editing model on a text, we test it by asking several questions, where each question targets a specific piece of knowledge found within that text. The x-axis represents the sequential order of knowledge within the original text, where position '1' corresponds to knowledge appearing at the very beginning, while position '>6' corresponds to knowledge located at the end. Our results show a consistent degradation in accuracy as the knowledge appears later in the text, which serves as the primary empirical evidence for the context reliance phenomenon. We have revised Section 3.1 and 3.2 to explicitly clarify details.
>
> **Comparision with Retrieval-based Methods:** COIN differs fundamentally from retrieval-based methods in two key aspects:
>
> * **Goal:** COIN aims to internalize knowledge by permanently updating the model's parameters, whereas retrieval methods strictly rely on external databases without altering the model itself.
>
> * **Utilization of Context: COIN leverages the original context only during the editing phase** to help the model learn the facts robustly. Crucially, at inference time, the edited model can recall this knowledge directly from its weights without the original context. In contrast, retrieval methods are entirely dependent on retrieving and providing the context at inference time for every query.
>
> **(W2) On the editing procedure in Figure 6**
>
> **Inputs/Outputs during editing:** The input is the unstructured text sequence containing the new knowledge (e.g., the 'Text' examples in Table 4 and 5). During editing, the model generates next-token probability distributions. **Uniquely in our framework, for each target token, the model produces two distributions, one conditioned on the full preceding text (global context) and one on a short sliding window (local context).** The context alignment loss forces the local distribution to match the global one, while the standard NTP loss optimizes the global prediction. When evaluating the edited model, we simply input the test questions (such as the "Completions" and "QAs" in Table 4 and 5) to the model, without requiring any additional context or external modules. We have polished Section 4 (Method) to make it more readable and easier to understand.
>
> **Consistency Objective:** **It precisely constrains the output vectors of the edited MLP layer, rather than the model's final output distribution.** We input a set of pre-computed irrelevant keys ($K_0$, sampled from a general corpus like Wikipedia) into the MLP. The loss minimizes the distance between the edited layer's output ($WK_0$) and the original layer's output ($W_0K_0$), ensuring that the module’s behavior on unrelated inputs remains mathematically identical to the pre-edited state.
>
> **(W3) On preserving unrelated knowledge**
>
> **To evaluate COIN on preserving model's general capabilities and unrelated knowledge, we conducted an experiment where we evaluated the edited models on GLUE benchmark after batch editing of 256 instances, shown in Figure 8 (right panel)**. The results show that COIN maintains high performance on these general tasks. This demonstrates that COIN successfully preserves the model's general capabilities and unrelated knowledge.

---

> ### Author Response · Authors · 2025-11-26
> **Response to Reviewer 8hZB (2/2)**
>
> **(W4) On computational cost**
>
> Thank you for your thoughtful comment. We agree that COIN introduces auxiliary computation, which we consider a worthwhile trade-off for its improvements in editing. The additional cost has two distinct sources:
>
> * **Context Alignment Loss:** This is the primary source of overhead. It requires a second forward pass to align predictions from global versus local contexts, increasing the per-step computational cost compared to a standard FT step.
>
> * **Knowledge Consistency Loss:** This loss has a more nuanced impact. The calculation per step is negligible, as it relies on a pre-computed matrix. However, by regularizing the model to preserve existing knowledge, it can increase the total number of training steps required for the edit to converge.
>
> We are committed to improving efficiency in future work by exploring optimizations such as more selective alignment strategies, as we note in Appendix B (Limitations & Future Discussion) under "Advanced Alignment Strategies."
>
> **(Q1) On the comparisons in Table 3**
>
> Thank you for the suggestion. **We note that the FT baseline already included in Table 3 is applicable to both structured and unstructured editing. For completeness, we have also added the results for AnyEdit and UnKE to Table 3 as requested, as shown below.** These additional comparisons demonstrate that COIN significantly outperforms other unstructured editing methods on structured knowledge editing tasks. We have updated the Table 3 to reflect these results.
>
> | Llama3-8B | Avg.  | 2-hops | 3-hops | 4-hops |
> | --------- | ----- | ------ | ------ | ------ |
> | AnyEdit   | 32.58 | 24.38  | 41.87  | 30.86  |
> | UnKE      | 34.01 | 28.75  | 41.58  | 30.41  |
> | FT        | 71.66 | 76.30  | 69.78  | 67.35  |
> | COIN      | 75.89 | 80.00  | 76.97  | 67.81  |
>
> **(Q2) On broader insights**
>
> **Our primary contribution is the discovery and systematic study of context reliance. We believe this issue is not only confined to knowledge editing, but also a more general factor that limits the generalization capabilities of models in various other fine-tuning tasks.** This opens promising future research into robust, generalizable fine-tuning strategies across domains.

---

### Official Review · Reviewer_TmC2 · 2025-10-27

**Soundness:** 3
**Presentation:** 3
**Contribution:** 3
**Rating:** 6
**Confidence:** 3

**Summary:**

This paper investigates a fundamental limitation of the Next-Token Prediction (NTP) paradigm when applied to unstructured knowledge editing in large language models (LLMs). The authors identify and formalize a phenomenon they call context reliance, where edited knowledge becomes overly dependent on its preceding context during editing, leading to retrieval failure when that context is missing at inference. To address this, the paper introduces COIN (COntext-INdependent editing), a framework augmenting standard NTP training. Extensive experiments on AKEW and UnKEBench datasets (with LLaMA3-8B and Qwen2.5-7B) demonstrate that COIN substantially improves editing success (up to 23.6% ROUGE-F1 gain) and reduces performance drop across positional contexts by 45%. Theoretical analysis (Theorem 3.1) and ablation studies support the identified cause of context reliance.

**Strengths:**

1. The paper identifies a subtle yet impactful issue—context reliance—that had not been rigorously studied before in the model editing literature. This conceptual framing provides a fresh lens for understanding why next-token-based fine-tuning often fails to generalize edited knowledge.

2. The COIN framework extends NTP training with two intuitive regularization terms that have clear theoretical underpinnings: one encourages invariance to context window size, and the other prevents catastrophic forgetting. The use of KL alignment between global and local distributions is simple yet effective, and the analytical formulation of the knowledge consistency constraint demonstrates mathematical rigor.

3. The paper is clearly written and logically structured. And proposed COIN achieves consistent improvements over strong baselines.

**Weaknesses:**

1. COIN currently uses a fixed-size sliding window for defining “local context,” which the authors themselves note as a limitation. This may underperform for long, discourse-rich texts where context relevance varies.

2. While AKEW and UnKEBench are suitable, they primarily test single-fact retrieval. It remains unclear whether COIN improves reasoning tasks that require integrating multiple edited facts or narrative comprehension. Authors can Include an evaluation on multi-fact narrative editing (e.g., multi-hop MQuAKE reasoning chains or synthetic story edits) to demonstrate generalization to compositional inference.

3. Both alignment and consistency losses add auxiliary computation, yet the paper does not report training time, memory, or scalability.

**Questions:**

How sensitive is COIN’s performance to the choice of window size k and the trade-off hyperparameters (α, β)? Are these fixed across datasets or tuned per task?

Can COIN be combined with existing locate-then-edit methods (e.g., ROME/AlphaEdit) to yield hybrid improvements?

Have the authors examined whether COIN affects unrelated factual recall accuracy on general QA datasets like Natural Questions or TriviaQA?

---

> ### Author Response · Authors · 2025-11-26
> **Response to Reviewer TmC2 (1/2)**
>
> We sincerely appreciate your recognition of our work. We hope the following responses will effectively address your concerns. Please let us know if you have any further questions or require additional clarification.
>
> **(W1) On fixed-size sliding window**
>
> Thank you for your feedback. **The choice of a fixed-size sliding window was intentional. As our primary contribution is identifying and analyzing the context reliance phenomenon, we designed COIN as a simple yet effective framework to provide a clear and interpretable solution.** This straightforward approach allows us to directly validate the efficacy of our proposed context alignment strategy without introducing additional complexities from more advanced mechanisms. We agree that this is a simplification and acknowledge it as a key limitation in Appendix B (Limitations & Future Discussion). We believe that exploring more sophisticated, adaptive strategies for defining local context is a promising direction for future research.
>
> **(W2) On evaluating multi-hop task**
>
> **As shown in Table 3, we present a comprehensive evaluation on the MQUAKE multi-hop dataset**. The results demonstrate that NTP-based methods, and COIN in particular, significantly outperform traditional locate-then-edit approaches. For completeness, we have updated the results of AnyEdit and UnKE in the table. These results further demonstrate that COIN exhibits superior performance in editing structured knowledge compared to other unstructured editing methods.
>
> **(W3) On computational cost**
> We agree that COIN introduces auxiliary computation, which we consider a worthwhile trade-off for its improvements in editing. The additional cost has two distinct sources:
>
> * **Context Alignment Loss:** This is the primary source of overhead. It requires a second forward pass to align predictions from global versus local contexts, increasing the per-step computational cost compared to a standard FT step.
>
> * **Knowledge Consistency Loss:** This loss has a more nuanced impact. The calculation per step is negligible, as it relies on a pre-computed matrix. However, by regularizing the model to preserve existing knowledge, it can increase the total number of training steps required for the edit to converge.
>
> We are committed to improving efficiency in future work by exploring optimizations such as more selective alignment strategies, as we note in Appendix B (Limitations & Future Discussion) under "Advanced Alignment Strategies."

---

> > ### Author Response · Authors · 2025-11-26
> > **Response to Reviewer TmC2 (2/2)**
> >
> > **(Q1) On sensitivity analysis to hyperparameters $(k, \alpha, \beta)$**
> >
> > Thank you for your questions. **Regarding the window size $k$, our analysis in Appendix I.3 shows that performance is generally stable.** Although a smaller $k$ slightly favors recall, we attribute this to the fact that a tighter context constraint prevents the model from attending to distant cues, effectively forcing it to internalize the new knowledge into its parameters to satisfy the alignment objective. For all hyperparameters $(k, \alpha, \beta)$, our methodology was to tune them once per model, as described in Appendix G (Implementation Details). **This single, optimal configuration was then fixed and used across all test datasets.** We chose this approach to rigorously test our method's generalization capabilities, showing that it performs well without needing to be re-tuned for every new task.
> >
> > **(Q2) On combination with other methods**
> >
> > Thank you for your question. **Although combine COIN with existing locate-then-edit methods would be difficult, but can be effectively applied to other NTP-based methods.** As noted in [1], these locate-then-edit methods typically target the hidden state of a single token. This approach is often insufficient for unstructured knowledge, which is inherently more complex than triplets. An alternative strategy is to decompose text into triplets and then apply those methods, as illustrated in the second paragraph of Section 1 (Introduction). However, this decomposition process breaks the connections in contexts. Applying alignment in this scenario would be counterproductive, as its purpose is to align local context with the global, and that meaningful global context has already been destroyed. **However, to demonstrate flexibility, we have conducted an experiment applying the context alignment loss to the LoRA framework.** The results below indicate that this integration yields a significant performance improvement over the standard LoRA baseline.
> >
> > | Llama3-8B      | Prec. | Rec.  | F1    | BERT  |
> > | -------------- | ----- | ----- | ----- | ----- |
> > | LoRA           | 48.52 | 51.57 | 46.64 | 64.46 |
> > | LoRA+Alignment | 52.72 | 74.23 | 55.28 | 71.54 |
> >
> > [1] AnyEdit: Edit Any Knowledge Encoded in Language Models, ICML'25
> >
> > **(Q3) On impact to unrelated fact**
> >
> > **To evaluate COIN on preserving model's general capabilities and unrelated knowledge, we conducted an experiment where we evaluated the edited models on GLUE benchmark after batch editing of 256 instances, shown in Figure 8 (right panel)**. The results show that COIN maintains high performance on these general tasks. This demonstrates that COIN successfully preserves the model's general capabilities and unrelated knowledge.

---

### Official Review · Reviewer_AES3 · 2025-10-30

**Soundness:** 2
**Presentation:** 3
**Contribution:** 3
**Rating:** 4
**Confidence:** 3

**Summary:**

The paper studies unstructured knowledge editing by sticking with the most “native” training objective—next-token prediction (NTP) over full text—rather than converting facts into triples or synthetic QA. The authors identify a robust failure mode they call context reliance: when the edited fact appears later in a paragraph, performance drops sharply; and if the query at test time omits the preceding context seen during editing, the model often fails to retrieve the fact. They show this empirically and offer a simple one-layer, one-step GD argument explaining why the learned mapping can hinge on a specific pair of context tokens. Building on this diagnosis, they propose COIN, adding a context-alignment loss to match predictions under full vs sliding-window context, and a knowledge-consistency loss to keep behavior stable on unrelated inputs.

**Strengths:**

- The paper clearly identifies and substantiates context reliance as the central failure mode of NTP-based unstructured editing—showing that edited knowledge becomes entangled with preceding context and collapses once that context is removed—offering a precise, empirically grounded diagnosis of why full-text fine-tuning often fails to generalize. This finding can contribute to the community.

- COIN’s two regularizers are easy to bolt onto standard NTP editing, and they directly target the identified gap (train with global context vs test with local context). The method section is straightforward.

- The experiments are comprehensive: on AKEW/UnKEBench, COIN achieves significant gains over the strongest baseline in terms of BERT/ROUGE-F1; on MQUAKE (multi-hop), it also substantially outperforms ROME, MEMIT, and AlphaEdit, demonstrating the superiority of the proposed method.

**Weaknesses:**

- The theoretical analysis is built on an extremely simplified setting—a single-layer Transformer, a single gradient-descent update, and an attention pattern dominated by just two tokens (p and q). While this abstraction is useful for illustrating how context reliance can emerge, it does not capture the dynamics of deeper, multi-head, multi-step training typical of actual LLMs.

- The details of experiment, such as sampling strategy and numerical stability of the covariance-style objective aren’t reported in enough detail.

- The main text largely focuses on the strengths and successful results of COIN. Important limitations—including the bounded theoretical analysis and the impact (if any) on language generation models beyond the tested scale—are acknowledged only in appendices or passing remarks.

- Limited Baseline Diversity: Most empirical comparisons focus on variants of NTP-based editing or classic baselines.

**Questions:**

1. Can the authors clarify precisely how local context windows $k$ are defined/selected, and how robust the approach is against variable-length or semantically structured local contexts? Would more adaptive windows improve performance or efficiency?

2. Given that the theoretical analysis (Theorem C.1) applies under strong simplifying assumptions, might the context reliance phenomenon be weaker/stronger in multi-layer, multi-step training scenarios? Did you observe any qualitative mismatches between theorem predictions and empirical findings?

3. The paper briefly mentions meta-learning editors such as MEND and memory-based approaches, but it remains unclear how these methods would behave under the same unstructured NTP setting. Have the authors considered, even qualitatively, whether such approaches exhibit similar context-reliance effects?

4. For the knowledge consistency loss, does increasing $|\mathbf{K}_0|$ (number of sampled keys) materially affect model collapse or knowledge retention? Additionally, does this approach generalize to "unstructured" or ambiguous keys, such as those found in natural dialogues?

---

> ### Author Response · Authors · 2025-11-26
> **Response to Reviewer AES3 (1/2)**
>
> Thank you for your thoughtful review. We hope the following points will clarify and resolve your concerns. Please let us know if you have any further questions or require additional clarification.
>
> **(W1) On the scope and purpose of the theoretical analysis**
>
> Thank you for recognizing the utility of our theoretical abstraction. Analyzing the full optimization dynamics of deep models is inherently complex and mathematically intractable. Consequently, **the single-layer, single-head attention setup is the standard and predominant setting adopted by the vast majority of theoretical work on Transformer training dynamics [1, 2, 3].** We followed this established methodology to ensure the analysis remains mathematically tractable. Our analysis aims to provide a clear, provable intuition for the origin of the context reliance phenomenon. We explicitly acknowledge this scope and highlight the extension to general settings as a key direction for future research in Appendix B (Limitations & Future Discussion).
>
> [1] Transformers learn in-context by gradient descent, ICML'23
>
> [2] Scan and Snap: Understanding Training Dynamics and Token Composition in 1-layer Transformer, NeurIPS'23
>
> [3] Transformers Provably Solve Parity Efficiently with Chain of Thought, ICLR'25
>
> **(W2 and Q4) On experimental details and stability analysis of $|K_0|$**
>
> We adopted the sampling strategy from [1], utilizing 100,000 key vectors from Wikipedia texts. **As Wikipedia comprises diverse unstructured text, this ensures our consistency constraint generalizes effectively to the ambiguous, natural language inputs found in real-world scenarios.** To verify numerical stability, we conducted a stability analysis varying $|K_0|$ from 10,000 to 100,000 on the first 300 cases from the AKEW dataset. **Performance remained consistent and nearly identical across all settings, confirming that our sampling provides a robust approximation of irrelevant knowledge.**
>
> [4] Locating and Editing Factual Associations in GPT, NeurIPS'22
>
> | $\|K_0\|$ | Prec. | Rec.  | F1    | BERT  |
> | ---------- | ----- | ----- | ----- | ----- |
> | 10,000     | 60.15 | 66.80 | 59.90 | 72.21 |
> | 25,000     | 60.15 | 66.80 | 59.90 | 72.18 |
> | 50,000     | 60.16 | 66.77 | 59.92 | 72.18 |
> | 75,000     | 60.21 | 66.85 | 59.99 | 72.24 |
> | 100,000    | 60.12 | 66.72 | 59.90 | 72.21 |
>
> **(W3) On displaying limitaions**
>
> Thank you for your valuable feedback. We clarify that the primary focus of our paper is the identification and thorough analysis of the context reliance phenomenon, a fundamental issue we believe is prevalent in unstructured knowledge editing. **We proposed COIN as a simple yet effective initial framework to demonstrate that this problem can be mitigated. The placement of the limitations section in the appendix was necessitated by strict page constraints, rather than an intent to hide potential drawbacks.** On the contrary, we strongly welcome discussions regarding limitations and future work, as they are essential for advancing this field.
>
> **(W4) On baseline diversity**
>
> Our evaluation concentrates on representative of the NTP-based methods (FT, LoRA, AdaLoRA), SOTA unstructured editors (UnKE, AnyEdit), and structured methods (ROME, MEMIT, AlphaEdit). **Since our primary contribution is identifying context reliance as a fundamental limitation of the NTP paradigm, demonstrating this phenomenon across this diverse set of methods provides sufficient empirical evidence to validate our claims.** Nevertheless, we appreciate your raising the question of whether other paradigms exhibit similar effects. We believe this is a meaningful direction and will continue to explore in future work.

---

> ### Author Response · Authors · 2025-11-26
> **Response to Reviewer AES3 (2/2)**
>
> **(Q1) On the selection of local context window size $k$ and sensitivity analysis**
>
> **In our framework, the local context is defined as a sliding window containing the $k$ tokens immediately preceding the current token being predicted. The selection of $k$ is empirically grounded in our sensitivity analysis (Appendix I.3 and Figure 12).** We observed that smaller window sizes (e.g., k=5 or 10) are more effective. We attribute this to the fact that a tighter context constraint prevents the model from attending to distant cues, effectively forcing it to internalize the new knowledge into its parameters to satisfy the alignment objective. Regarding adaptive window, we agree with your insight. As discussed in the Appendix B (Limitations & Future Discussion), we acknowledge that a fixed-window approach is a heuristic that may not optimally capture semantic boundaries in all cases. We agree that adaptive strategies, such as determining window size based on attention entropy or syntactic structure, could further enhance robustness. We appreciate this valuable suggestion and consider it a promising direction for our future work.
>
> **(Q2) On the gap between theory and empirical findings**
>
> Thank you for pointing this out. **Our theoretical analysis and empirical findings are consistent, both designed to demonstrate the existence of the context reliance phenomenon.** Our theoretical analysis in Theorem 3.1 serves as a proof of how context reliance can fundamentally emerge from the training process. This theoretical prediction is directly validated by our empirical findings, as illustrated in Figure 3. The experiment shows that the probability of a correct answer drops when the preceding context is removed and increases when it is included, which precisely validates our theorem. To make this alignment between our theory and experiments clearer, we have revised Section 3.2 (Context Reliance Phenomenon) to explicitly clarify details.
>
> **(Q3) On demonstration of MEND and memory-based approaches under the unstructured setting**
>
> Thank you for this insightful question. Our work primarily focuses on limitations of NTP paradigm for unstructured knowledge editing, especially parameter-modifying methods, so we did not include memory-based approaches in our analysis.**Regarding meta-learning editors like MEND, we hypothesize that they would exhibit similar context reliance issues.** Since these approaches train a hypernetwork to predict weight updates that minimize an objective on given edit. If the objective remains the standard NTP loss conditioned on the full text $P(target|context)$, the resulting weight updates may show the same flaw. **Unless the training objective specifically simulates context absence, the resulting edits will likely retain the dependency on the context.**

---

### Official Review · Reviewer_awnm · 2025-10-31

**Soundness:** 2
**Presentation:** 3
**Contribution:** 2
**Rating:** 4
**Confidence:** 4

**Summary:**

The paper examines unstructured knowledge editing in the next-token prediction (NTP) setting and observes that models can become reliant on the preceding context used during editing; consequently, the edited knowledge is harder to retrieve when queried without that context. To address this, the authors propose COIN, which augments the standard editing loss with a context-alignment objective that encourages the model’s predictions under a global window to match those under a local window. The approach is evaluated on several unstructured editing benchmarks.

**Strengths:**

1. The paper targets a meaningful and timely issue in unstructured knowledge editing under NTP, with clear implications for reliability and deployment.

2. The finding that edited knowledge retrieval depends on the preceding context is interesting and practically important; the proposed mitigation idea is reasonable.

3. The paper is generally well written and easy to follow; figures/tables and section flow make the narrative accessible.

**Weaknesses:**

1. Some setup details need elaboration. namely,  Section 3.3’s two mitigation strategies are insufficiently specified. It remains ambiguous how they are implemented in practice. Similarly, for splitting/paraphrasing, it is unclear whether the order of knowledge is permuted to control for position-dependent difficulty. Concrete examples (with before/after text) would help disambiguate the procedure.

2. The central “alignment loss” idea, which aims to remove global context impacts, does not appear new in my opinion. It has been explored in the long-context learning/understanding literature [1]. The authors should make it clearer on the connection and uniqueness of the solution proposed in this paper.

3. The design of consistency loss solution looks suboptimal. As mentioned by the authors, the concept of "unrelated knowledge" $K_0$ as in ROME/MEMIT/AlphaEdit. However, the the paper opts for a generic regularization on $W_0$ rather than a more structure-aware constraint as in AlphaEdit.

4. As claimed by the authors, the core issue identified in this work is inherent to the NTP training paradigm. In addition, the proposed alignment loss should be tested across diverse NTP-based editors (fine-tuning, LoRA, ROME/MEMIT) to demonstrate generality and systematic gains, rather than a specific method.

5. The problem appears related to known overfitting phenomena in editing, which has been widely studied in the literature [2, 3, 4, 5], but these links were not discussed clearly.

[1] What Is Wrong with Perplexity for Long-Context Language Modeling? 2024.

[2] Neighboring Perturbations of Knowledge Editing on Large Language Models, 2024.

[3] Uncovering Overfitting in Large Language Model Editing, 2025.

[4] Revealing and Mitigating Over-Attention in Knowledge Editing, 2025.

[5] Mitigating Heterogeneous Token Overfitting in LLM Knowledge Editing, 2025.

**Questions:**

Please see my comments in the weakness sections.

---

> ### Author Response · Authors · 2025-11-26
> **Response to Reviewer awnm (1/2)**
>
> Thank you for acknowledging the contributions of our work. We hope the following points will clarify and resolve your concerns. Please let us know if you have any further questions or require additional clarification.
>
> **(W1) On setup details of mitigation strategies and response to whether the order of knowledge is permuted**
>
> Thank you for your suggestion. **We clarify that neither strategy is intended to permute the order of knowledge.** The specific implementations are as follows:
>
> * **Knowledge Splitting:** We split the long-form text into independent short sentences based on sentence boundaries to serve as training samples. All of these sentences are served as training samples along with the original text. **This design forces the model to predict knowledge even when context is absent, thereby reducing the model's reliance on preceding context.**
>
> * **Paraphrasing:** We utilized LLMs to rewrite the text with diverse syntactic structures while preserving the original semantics and knowledge points. **This strategy aimed to test whether diverse phrasing could effectively break the model's reliance on context.**
>
> As requested, we have added concrete text examples below to illustrate these procedures.
>
> ```
> Original Text:
> Francisco Serrano, 1st Duke of la Torre, was a Spanish military officer and politician who rose to prominence during the 19th century. After serving in various military campaigns, Serrano found employment in Rome, where he worked as a diplomat and ambassador for the Spanish government. His skills and experience in diplomacy proved valuable in maintaining strong relations between Spain and other European countries.
>
> After Knowledge Splitting:
> 1. Francisco Serrano, 1st Duke of la Torre, was a Spanish military officer and politician who rose to prominence during the 19th century.
> 2. After serving in various military campaigns, Serrano found employment in Rome, where he worked as a diplomat and ambassador for the Spanish government.
> 3. His skills and experience in diplomacy proved valuable in maintaining strong relations between Spain and other European countries.
>
> After Paraphrasing:
> Francisco Serrano, the 1st Duke of la Torre, was a notable Spanish military leader and statesman who gained recognition in the 19th century. Following his participation in multiple military campaigns, he relocated to Rome, where he served as a diplomat and ambassador for Spain. His diplomatic expertise played a key role in strengthening Spain's ties with other European nations.
> ```
>
> **(W2) On comparision between context alignment loss and works in long-context modeling**
>
> Thank you for pointing out this work. **Essentially, [1] aims to exploit the gap between global and local contexts to enhance long-context capabilities, whereas we aim to eliminate it to enhance inference without context.**
>
> * [1] focuses on long-context modeling, where this gap is treated as a positive signal of effective retrieval or in-context learning. They aim to exploit this gap to reinforce the model's ability to extract information from the context.
>
> * In contrast, in the domain of knowledge editing, we identify this gap as a failure mode where the model relies on the preceding text rather than storing the new knowledge in its parameters. We aim to eliminate this gap so that the edited model can recall knowledge without preceding context during inference.
>
> [1] What Is Wrong with Perplexity for Long-Context Language Modeling?, ICLR'25
>
> **(W3) On comparision between knowledge consistency loss and null space projection in AlphaEdit**
>
> Thank you for your comment. On the one hand, **our objective is identical to AlphaEdit**, both aim to satisfy $WK_0=W_0K_0$ to preserve unrelated knowledge. While AlphaEdit enforces this via null-space projection, we formulate it as a regularization term. On the other hand, **our design is crucial for compatibility with our context alignment loss**. Since context alignment requires iterative gradient updates, the projection step is hard to compatible with this learning process. By formulating consistency as a loss term, COIN allows the optimizer to jointly balance knowledge injection and preservation via gradient descent.

---

> ### Author Response · Authors · 2025-11-26
> **Response to Reviewer awnm (2/2)**
>
> **(W4) On combination with other NTP methods**
>
> Thank you for this valuable feedback. **Our proposed alignment strategy is compatible with other NTP-based methods like LoRA, demonstrating its generalizability.** To validate this, we conducted additional experiments integrating the context alignment loss into LoRA on the AKEW benchmark, and the results below indicate that this integration yields significant performance improvements over the standard LoRA baseline. These findings confirm that our alignment loss can serve as a plug-in module, effectively enhancing the robustness of NTP-based editing methods.
>
> | Llama3-8B      | Prec. | Rec.  | F1    | BERT  |
> | -------------- | ----- | ----- | ----- | ----- |
> | LoRA           | 48.52 | 51.57 | 46.64 | 64.46 |
> | LoRA+Alignment | 52.72 | 74.23 | 55.28 | 71.54 |
>
> **(W5) On discussion between context reliance and overfitting phenomena in editing**
>
> Thank you for the insightful suggestion to connect context reliance with overfitting in editing. Viewing our work through this lens offers a valuable perspective. We clarify the relationship and differences as follows:
>
> * **Dependency Source:** In structured knowledge editing, **overfitting typically occurs when model binds target to specific subject or relation tokens [3, 4].** This causes model to generate target whenever these tokens appear, regardless of the context. In contrast, **context reliance in unstructured NTP-based editing arises because the model binds the target to the entire preceding context.** This context often contains irrelevant noise or narrative elements, and model falsely learns these contextual patterns as necessary cues.
>
> * **Generalization Outcome: Overfitting can be viewed as a kind of over-generalization.** The model outputs the edited knowledge in inappropriate scenarios, leading to damage on neighboring knowledge [2] or heterogeneous overfitting on specific tokens [5]. **In contrast, context reliance is a kind of under-generalization.** The model fails to recall knowledge when the specific training context is absent, limiting the knowledge's availability to practical inference.
>
> [2] Neighboring Perturbations of Knowledge Editing on Large Language Models, ICML'24
>
> [3] Uncovering Overfitting in Large Language Model Editing, ICLR'25
>
> [4] Revealing and Mitigating Over-Attention in Knowledge Editing, ICLR'25
>
> [5] Mitigating Heterogeneous Token Overfitting in LLM Knowledge Editing, ICML'25

---

### Author Response · Authors · 2025-12-03
**General Response to Area Chair (Part 2/2): Resolution of Concerns**

### **Resolution of Concerns**

During the rebuttal, we engaged extensively with all reviewers, providing theoretical clarifications, additional experimental results, and expanded baselines. We have incorporated these key updates into the revised manuscript, where they are highlighted for reference. We believe the following actions have effectively resolved the raised concerns:

1. **Reviewer awnm**

* **Concerns:** Clarification on mitigation strategies,  distinction from prior works, and compatibility with other methods.

* **Action:**

  * **Clarification:** We **detailed the implementation** of knowledge splitting and paraphrasing, providing concrete text examples to illustrate how these strategies fail to break context reliance.

  * **Distinction:** We **clarify the connection and uniqueness of COIN** against study in the long-context learning as required. We further **distinguish context reliance from traditional overfitting**, noting that while overfitting represents over-generalization to specific tokens, context reliance is a form of under-generalization. Finally, our consistency objective differs from AlphaEdit by utilizing a regularization term, which is **crucial for compatibility** with the dynamic gradient updates required by our context alignment loss.

  * **Compatibility:** We **conducted additional experiments integrating our alignment loss into LoRA** on the AKEW benchmark. Results showed significant performance improvements, demonstrating that **our alignment strategy serves as a plug-in module compatible with other NTP-based editors**.

2. **Reviewer AES3**

* **Concerns:** Simplified theoretical analysis, stability of the covariance approximation, and baseline discussion.

* **Action:**

  * **Theoretical Scope:** We justified the single-layer setup as standard practice in theoretical deep learning analysis to keep derivations tractable while providing valid intuition. Crucially, we highlighted that **our empirical findings (Figure 3) on large-scale models perfectly align with the theoretical predictions**, validating the theory's practical relevance.

  * **Stability Analysis:** We conducted a sensitivity analysis on the number of sampled keys for the consistency loss. Results confirmed that **performance remains stable and consistent across varying sample sizes (from 10k to 100k).**

  * **Baseline Discussion:** We clarified that **we focus on the NTP paradigm**. We hypothesized and explained why meta-learning approaches (like MEND) would likely suffer similar context reliance if their training objective remains conditioned on the full text without alignment constraints.

3. **Reviewer TmC2**

* **Concerns:** Fixed-size sliding window limitation, generalization to multi-hop reasoning, and computational cost.

* **Action:**

  * **Multi-hop Reasoning:** We evaluated COIN on the MQUAKE dataset. Results in Table 3 demonstrate that COIN significantly **outperforms traditional locate-then-edit approaches and other unstructured editors** on multi-hop tasks, proving robust generalization.

  * **Sensitivity Analysis:** We performed an ablation study on the window size, and the analysis in Appendix I.3 shows **COIN is generally stable**. All hyperparameters $(k, \alpha, \beta)$ was same for all tasks per model, as described in Appendix G, showing that **it performs well without needing to be re-tuned for every new task**.

  * **Cost Justification:** We acknowledged the auxiliary computation but argued it is a worthwhile trade-off for the substantial gains (up to 23.6%) in editing reliability.

4. **Reviewer 8hZB**

* **Concerns:** Impact of knowledge position, comparison with structured/retrieval methods, and preservation of general capabilities.

* **Action:**

  * **Position Analysis:** We clarified that the impact of knowledge position was **already explicitly analyzed** in Figure 2 and Section 3.1 of our original submission, and we provided further explanation to resolve this confusion.

  * **Comparisons:** As requested, we updated Table 3 to **include comparisons with unstructured editors** AnyEdit and UnKE. COIN significantly outperforms both on structured knowledge editing tasks, highlighting the versatility of our approach.

  * **General Capabilities:** We have evaluated the edited models on the GLUE benchmark (Figure 8). The results confirm that **COIN successfully preserves the model's general language understanding capabilities** compared to baselines.

We respectfully hope this summary aids in your final assessment.

Best regards,

The Authors

---

### Author Response · Authors · 2025-12-03
**General Response to Area Chair (Part 1/2): Summary of Paper & Summary of Strengths**

Dear Area Chair,

Thank you for overseeing the review process of our submission. To assist in your final assessment, we provide a comprehensive summary of the rebuttal process between the reviewers and us.

### **Summary of Paper**

**Our work theoretically and empirically identifies Context Reliance, a critical yet previously overlooked failure mode in unstructured knowledge editing** under the Next-Token Prediction (NTP) paradigm, where models fail to recall edited knowledge when the specific training context is absent. To address this, **we propose COIN, a simple yet effective framework** introducing a **Context Alignment Loss** to decouple knowledge from its context and internalize new knowledge, and a **Knowledge Consistency Loss** to prevent model collapse. COIN achieves state-of-the-art results, **outperforming strong baselines by 23.6% in editing success rates** on the AKEW and UnKEBench benchmarks.

### **Summary of Strengths**

All reviewers acknowledged the significance of our work. We summarize the strengths below:

1. **Significance of Context Reliance:** Reviewers consistently highlighted the significance and novelty of identifying the Context Reliance. As illustrated by reviewers, **this is a "significant" [8hZB], "meaningful and timely issue" [awnm] that "had not been rigorously studied before" [TmC2]. Our finding is "interesting and practically important" [awnm] and can "contribute to the community" [AES3], "offering a precise, empirically grounded diagnosis of why full-text fine-tuning often fails to generalize" [AES3].**

2. **Simple and Effective Framework:** The proposed COIN framework was highlighted as **"simple yet effective" [TmC2, 8hZB]** and **"intuitive" [TmC2]**. Reviewers appreciated that the regularizers are **"easy to bolt onto standard NTP editing" [AES3]** while directly targeting the identified gap between training and inference contexts.

3. **Comprehensive Evaluation:** Reviewers noted that COIN achieves **"consistent improvements" [TmC2] and "significant gains" [AES3] over strong baselines** across multiple benchmarks.

---

### Meta-Review · Area_Chair_9Ffd · 2026-01-07

**Summary:**

This paper studies a fundamental limitation of unstructured knowledge editing under the next-token prediction (NTP) paradigm, identifying a failure mode termed context reliance, where edited knowledge becomes entangled with the specific preceding context used during editing and cannot be reliably retrieved when that context is absent at inference time. The authors provide both empirical evidence across multiple benchmarks (AKEW, UnKEBench, MQUAKE) and a simplified theoretical analysis to explain how this phenomenon arises naturally from NTP optimization.

To mitigate this issue, the paper proposes COIN, which augments standard NTP-based editing with (1) a context alignment loss enforcing consistency between predictions under global and local context windows, and (2) a knowledge consistency loss to preserve unrelated knowledge. Experiments show consistent and often substantial gains over fine-tuning and strong unstructured editing baselines, as well as competitive or superior performance compared to structured editors in multi-hop settings.

Across reviews, there is broad agreement that the problem formulation and diagnosis are important and timely, and that the empirical results demonstrate clear benefits of the proposed approach. The main points of contention concern (1) the novelty of the alignment idea relative to prior work on long-context learning and overfitting in editing, (2) the simplicity and limited scope of the theoretical analysis, (3) clarity of experimental details and training procedures, and (4) computational cost and generality beyond NTP-based methods. The rebuttal provides extensive clarifications, additional experiments, and conceptual positioning that address most of these concerns. Remaining disagreements are largely about conceptual framing and expectations of theoretical depth, rather than empirical validity.

**Reviewer Concerns:**

Concerns largely addressed by the rebuttal:

Insufficient methodological clarity (Reviewers awnm, 8hZB):
The rebuttal provides concrete, step-by-step explanations of the mitigation strategies (knowledge splitting, paraphrasing), detailed examples of before/after texts, and clearer descriptions of the training procedure and objectives (including what the consistency loss constrains). These additions substantially improve reproducibility and clarity.

Relationship to prior work on long-context learning and overfitting (Reviewer awnm):
The authors clearly distinguish COIN’s goal (eliminating global–local gaps to force parameter internalization) from long-context work that exploits such gaps, and explicitly position context reliance as a form of under-generalization, contrasting it with known overfitting phenomena. This framing satisfactorily situates the work within the broader literature.

Generality beyond a single editor (Reviewers awnm, TmC2):
Additional experiments integrating the alignment loss into LoRA demonstrate that COIN’s core idea is not tied to a single fine-tuning setup, supporting its role as a plug-in enhancement for NTP-based editors.

Hyperparameter sensitivity and window size choice (Reviewers AES3, TmC2):
New sensitivity analyses show stable performance across window sizes and sampled keys, with hyperparameters tuned once per model and reused across datasets, alleviating concerns about fragility.

Preservation of unrelated knowledge and model capabilities (Reviewers AES3, 8hZB):
Evaluation on GLUE after batch editing shows minimal degradation, addressing concerns about localization and catastrophic forgetting.

Computational overhead (Reviewers TmC2, 8hZB):
The rebuttal clarifies the sources of overhead (additional forward pass for alignment, slower convergence due to regularization) and argues that the cost is moderate and justified by performance gains, while acknowledging efficiency as future work.

Concerns partially addressed or still remains:

Theoretical depth and realism (Reviewer AES3):
While the authors appropriately justify the simplified theoretical setting as standard practice, the analysis remains illustrative rather than predictive for deep, multi-layer optimization dynamics. This limitation is acknowledged but not fully resolved.

Perceived novelty and conceptual overlap (Reviewer awnm):
Although distinctions from prior alignment and overfitting work are clarified, some reviewers may still view the core alignment idea as incremental rather than fundamentally new.

Skepticism toward NTP-based editing overall (Reviewer 8hZB):
Reviewer 8hZB remains unconvinced that COIN sufficiently advances the state of unstructured editing relative to simpler fine-tuning or alternative paradigms, despite additional comparisons and explanations. This appears to reflect a difference in research perspective rather than an unresolved technical flaw.

**Reviewer Scores:**

Reviewer awnm (Initial: 4, marginal reject):
Most concerns regarding clarity, novelty positioning, generality across methods, and relation to overfitting were directly addressed with new experiments and detailed explanations.
Likely updated score: 5–6 (borderline to weak accept)

Reviewer AES3 (Initial: 4, marginal reject):
The rebuttal resolves questions about experimental details, window selection, stability, and consistency loss behavior, while acknowledging theoretical limitations.
Likely updated score: 5–6 (borderline to weak accept)

Reviewer TmC2 (Initial: 6, marginal accept):
Additional analyses on hyperparameter sensitivity, multi-hop evaluation, and preservation of general capabilities strengthen an already positive assessment.
Likely updated score: 6–7 (steady or slightly stronger accept)

Reviewer 8hZB (Initial: 2, reject):
While many technical clarifications and additional experiments were provided, the reviewer’s fundamental skepticism toward the paradigm and framing likely remains.
Likely updated score: 3 (still below threshold)

---

### Decision · Program_Chairs · 2026-01-26

Reject